# META-VALUE LEARNING: A GENERAL FRAMEWORK FOR LEARNING WITH LEARNING AWARENESS

## ABSTRACT

Gradient-based learning in multi-agent systems is difficult because the gradient derives from a first-order model which does not account for the interaction between agents' learning processes. LOLA (Foerster et al., 2018a) accounts for this by differentiating through one step of optimization. We propose to judge joint policies by their long-term prospects as measured by the meta-value, a discounted sum over the returns of future optimization iterates. We apply a form of $Q$-learning to the meta-game of optimization, in a way that avoids the need to explicitly represent the continuous action space of policy updates. The resulting method, MeVa, is consistent and far-sighted. We analyze the behavior of our method on a toy game and compare to prior work on repeated matrix games.

## 1 INTRODUCTION

Multi-agent reinforcement learning (Busoniu et al., 2008) has found success in two-player zero-sum games (Mnih et al., 2015; Silver et al., 2017), cooperative settings (Lauer, 2000; Matignon et al., 2007; Foerster et al., 2018b; Panait & Luke, 2005), and team-based mixed settings (Lowe et al., 2017). General-sum games, however, have proven to be a formidable challenge.

The classic example is the Prisoner's Dilemma, a matrix game in which two players must decide simultaneously whether to cooperate or defect. Both players prefer cooperate-cooperate over defect-defect, but unilaterally both players prefer to defect. The game becomes qualitatively different when it is infinitely repeated and players can see what their opponents did in previous rounds: this is known as the Iterated Prisoner's Dilemma (IPD). This gives players the opportunity to retaliate against defection, which allows cooperation with limited risk of exploitation. The most well-known retaliatory strategy is tit-for-tat (Axelrod & Hamilton, 1981), which cooperates initially and from then on copies its opponent. It was only recently discovered that there exist ZD-extortion policies (Press & Dyson, 2012) which force rational opponents to accept considerable losses. Such exortionate policies, however, perform worse against themselves than tit-for-tat does.

The problem of learning to coordinate in general-sum games has received considerable attention over the years (Busoniu et al., 2008; Gronauer & Diepold, 2022). The naive application of gradient descent (see §2.1) fails to find tit-for-tat on the IPD unless initialized sufficiently close to it (Foerster et al., 2018a). Instead, it converges to always-defect, to the detriment of both players. Several works approach the problem through modifications of the objective, e.g. to share reward (Baker, 2020), to explicitly encourage interaction (Jaques et al., 2019) or encourage fairness (Hughes et al., 2018). While these approaches may work, they could achieve cooperation for the wrong reasons. We instead are interested in achieving cooperation only where *self-interest* warrants it.

We take inspiration from the recent work Learning with Opponent-Learning Awareness (LOLA Foerster et al. (2018a;c)), the first general learning algorithm to find tit-for-tat on IPD. LOLA mitigates the problems of naive learning by extrapolating the optimization process into the future: it simulates a naive update and evaluates the objective at the resulting point. By differentiating through the extrapolation, LOLA effectively uses second-order information to account for the learning process. This allows for *opponent shaping*: intentionally influencing the opponent's learning process to our benefit. The term typically refers to dynamical exploitation of the opponent's learning process, but similar effects can be achieved through fixed policies with threats (e.g., tit-for-tat, ZD extortion).

A shortcoming of LOLA's approach is that it is inconsistent (Willi et al., 2022): the extrapolation process that it aims to influence does not match the true learning process. Moreover, it is not effectively able to extrapolate far into the future, which limits its ability to escape the basin of attraction of a poor solution. We make the following contributions:

- We propose to follow the gradient of the meta-value function, which is naturally consistent and readily accounts for longer-term and higher-order interactions. In §3 we demonstrate how to approximate this ideal.
- Unlike prior work, our approach is based on value learning and does not require policy gradients anywhere. However, we provide a variant (§4.1) that mitigates the downsides of bootstrapping, generally at the cost of requiring policy gradients on the inner game.
- Our $Q$-function is implemented implicitly through a state-value function $V$ (§3.3). We approximate the greedy (argmax) action as the gradient of $V$, sidestepping the need to explicitly represent the continuous meta-action space.
- We demonstrate the importance of looking far ahead in §5.1, and show qualitatively how the gradient of the meta-value leads to the Pareto-efficient solution regardless of initialization.
- In a tournament on repeated matrix games (§5.2), MeVa exhibits opponent-shaping behavior, including ZD-extortion (Press & Dyson, 2012) on the IPD, and dynamical exploitation on Iterated Matching Pennies.

Code is available at https://github.com/MetaValueLearning/MetaValueLearning.

## 2 BACKGROUND

We study $P$-player differentiable games $f : \mathbb{R}^{P \times N} \mapsto \mathbb{R}^P$ that map vectors of policies $x_i$ to vectors of expected returns $y_i$,

$$\begin{pmatrix} y_1 \\ \vdots \\ y_P \end{pmatrix} = f \begin{pmatrix} x_1 \\ \vdots \\ x_P \end{pmatrix},$$

or $y = f(x)$ for short. For simplicity, we assume $x_i \in \mathbb{R}^N$ for each player $i \in \{1, \ldots, P\}$ are real-valued parameter vectors that represent policies through some fixed parametric class (e.g. a lookup table or a neural network). We will often describe the game from the perspective of a single player $i$, and separate their objective $f_i(x) \in \mathbb{R}$ from those of the others $f_{-i}(x) \in \mathbb{R}^{P-1}$, or separate their parameters $x_i \in \mathbb{R}^N$ from those of the others $x_{-i} \in \mathbb{R}^{(P-1) \times N}$, as in $f(x_i, x_{-i})$. All of our experiments concern two-player games with $P = 2$.

### 2.1 NAIVE LEARNING

Under naive learning, agents $i$ simultaneously update their policies according to

$$x_i^{(t+1)} = x_i^{(t)} + \alpha \nabla_{x_i} f_i(x^{(t)}) \tag{1}$$

where $\alpha$ is a learning rate and $\nabla_{x_i}$ is the gradient with respect to player $i$'s parameters. We may write this in vector form as

$$x^{(t+1)} = x^{(t)} + \alpha \bar{\nabla}_x f(x^{(t)}),$$

where $\bar{\nabla}_x$ denotes differentiation of each $f_i$ with respect to the corresponding policy $x_i$.

Naive learning is the straightforward application of standard gradient descent which is popular when optimizing a single objective. However, the gradient derives from a local first-order model – it reflects change in the objective due to change in each of the individual parameters, all others held equal. Simultaneous updates depart from this model, and while they are effective in the single-objective case, they fail when different elements of $x$ optimize different objectives.

### 2.2 LOOKING AHEAD

A number of approaches in the literature aim to address this issue by "looking ahead", considering not just the current parameter values $x^{(t)}$ but also an extrapolation based on an imagined update.

They essentially replace the game $f$ with a surrogate game $\tilde{f}$ that evaluates $f$ after an imagined naive update with learning rate $\alpha$:

$$\tilde{f}(x) = f(x + \alpha\bar{\nabla}_x f(x)). \tag{2}$$

Concretely, Zhang & Lesser (2010) consider a surrogate that only extrapolates opponents:

$$\tilde{f}_i(x) = f_i(x_i, x_{-i} + \alpha\bar{\nabla}_{x_{-i}} f_{-i}(x)). \tag{3}$$

Here $\bar{\nabla}_{x_{-i}} f_{-i}(x)$ stands for the naive gradients of the opponents $-i$. In computing the associated updates $\bar{\nabla}_x \tilde{f}(x)$, the authors considered the imagined update $\bar{\nabla}_x f(x)$ constant with respect to $x$. LOLA (Foerster et al., 2018a;c) introduced the idea of *differentiating through* $\bar{\nabla}_x f(x)$, thus incorporating crucial second-order information that accounts for interactions between the learners.

Unfortunately, this surrogate trades one assumption for another: while it no longer assumes opponents to stand still, it now assumes them to update according to naive learning. Foerster et al. (2018a) also proposed Higher-Order LOLA (HOLA), where HOLA0 assumes opponents are fixed, HOLA1 assumes opponents are naive learners, HOLA2 assumes opponents use LOLA, and so on. There is nevertheless always an inconsistency between the agent's assumption and the opponent's true learning process. Moreover, by imagining only the updates of the opponents, these approaches can be said to assume *themselves* to stand still. In this sense, LOLA and several of its derivatives are *inconsistent*.

To avoid both sources of inconsistency, we should like to use a consistent surrogate, such as

$$\tilde{f}^\star(x) = f(x + \alpha\bar{\nabla}_x \tilde{f}^\star(x)), \tag{4}$$

which assumes all players follow the naive gradient on $\tilde{f}^\star$ itself. When all players follow the naive gradient on $\tilde{f}^\star$, the extrapolation is consistent with the true learning process. Unfortunately, (4) is an implicit equation; it is unclear how to obtain the associated update $\bar{\nabla}_x \tilde{f}^\star(x)$.

COLA (Willi et al., 2022) solves such an implicit equation in the case where only the opponents are extrapolated, but the approach is generally applicable. In essence, they approximate the gradient $\bar{\nabla}_x \tilde{f}^\star(x)$ by a model $\hat{g}(x; \theta)$. The model is trained to satisfy

$$\hat{g}(x; \theta) = \bar{\nabla}_x f(x + \alpha\hat{g}(x; \theta)) \tag{5}$$

by minimizing the squared error between both sides. When the equation is tight, $\hat{g}(x; \theta) = \bar{\nabla}_x \tilde{f}^\star(x)$, providing access to the gradient of the consistent surrogate (4).

## 2.3 GOING META

Several approaches consider optimization as a meta-game. In its simplest form, the meta-game is a repeated game that has $f$ as its stage game. Thus the meta-state space is that of joint policies $x \in \mathbb{R}^{P \times N}$, and each player's meta-action $x_i' \in \mathbb{R}^N$ is the policy they wish to play next. Meta-actions $x_i'$ are chosen according to meta-policies $\pi_i$, resulting in joint meta-action $x' \sim \pi(\cdot \mid x)$ according to joint meta-policy $\pi(x' \mid x) = \prod_i \pi_i(x_i' \mid x)$, and joint meta-rewards $f(x')$ given by the stage game. The expected meta-return from a given meta-state $x^{(t)}$ is given by the meta-value

$$V_i^\pi(x^{(t)}) = \mathbb{E}_{x^{(>t)} \sim \pi} \sum_{\tau=t}^\infty \gamma^{\tau-t} f_i(x^{(\tau)}) = f_i(x^{(t)}) + \gamma \mathbb{E}_{x^{(t+1)} \sim \pi} V_i^\pi(x^{(t+1)}). \tag{6}$$

In this context, gradient methods like naive learning and LOLA can be seen as deterministic meta-policies, although they become stochastic when policy gradients are involved, and non-Markov when path-dependent information like momentum is used.

Meta-PG (Al-Shedivat et al., 2018) was the first to consider such a meta-game, applying policy gradients to find initializations $x_i$ that maximize $V_i^\pi$, with $\pi$ assumed to be naive learning on $f$. Meta-MAPG (Kim et al., 2021) tailor Meta-PG to multi-agent learning, taking the learning process of other agents into account. However, Meta-MAPG (like Meta-PG) assumes all agents use naive learning, and hence is inconsistent like LOLA.

M-FOS (Lu et al., 2022) considers a partially observable meta-game, thus allowing for the scenario in which opponent policies are not directly observable. M-FOS trains parametric meta-policies $\pi_i(\cdots; \theta_i)$ to maximize $V_i^\pi$ using policy gradients. However, the move to arbitrary meta-policies, away from gradient methods, discards the gradual dynamics that are characteristic of learning. As such, M-FOS does not learn to *learn* with learning awareness so much as learn to *act* with learning awareness. Indeed, M-FOS uses arbitrarily fast policy changes to derail naive and LOLA learners.

## 3 META-VALUE LEARNING

We now describe our method. First, we propose the use of the meta-value function as a surrogate game. Next, we demonstrate how to estimate the gradient of the meta-value using standard reinforcement learning techniques. Finally, we discuss how the proposed meta-policy relates to one that is (locally) greedy in the $Q$-values.

### 3.1 THE META-VALUE FUNCTION

In order to fully account for the downstream effects of our meta-actions, we propose to follow the gradient of the meta-value function of (6):

$$x_i^{(t+1)} = x_i^{(t)} + \alpha \nabla_{x_i} V_i(x^{(t)}). \tag{7}$$

Like the popular surrogate (3), it looks ahead in optimization time, but it does so in a way that is consistent with the true optimization process $x^{(t)}$ and naturally covers multiple steps.

When all players follow this meta-policy, the transition is deterministic and we can write

$$V(x) = f(x) + \gamma V(x') \quad \text{with} \quad x' = x + \alpha \bar{\nabla}_x V(x), \tag{8}$$

which is consistent like (4), and provides a natural way to look further into the future through the discount rate $\gamma$. It is however implicit, so we cannot directly access the gradients $\nabla_{x_i} V_i$ used in (7).

### 3.2 LEARNING META-VALUES

We will take a similar approach as Willi et al. (2022) and approximate our implicit surrogate (8) with a model. Specifically, we propose to learn a model $\hat{V}_i(x_i; \theta_i)$ parameterized by $\theta_i$ that approximates the values $\hat{V}_i \approx V_i$, and use its gradients $\nabla_{x_i} \hat{V}_i \approx \nabla_{x_i} V_i$ instead. Thus rather than emitting entire gradients (as COLA does) or emitting entire policies (as M-FOS does), we model scalars, and estimate the gradient of the scalar by the gradient of the estimated scalar. The resulting algorithm is related to Value-Gradient Learning (Fairbank & Alonso, 2012), but we do not directly enforce a Bellman equation on the gradients $\nabla_{x_i} \hat{V}_i$.

In essence, the learning process follows a nested loop (see Algorithms 1 & 2). In the inner loop, we collect a (finite) policy optimization trajectory according to

$$x_i^{(t+1)} = x_i^{(t)} + \alpha \nabla_{x_i} \hat{V}_i(x^{(t)}; \theta) \quad \text{and} \quad x_{-i}^{(t+1)} \sim \pi_{-i}(\,\cdot\mid x^{(t)}). \tag{9}$$

Then in the outer loop, we train $\hat{V}$ by minimizing the total TD error

$$\sum_t (\delta_i^{(t)})^2 \quad \text{where} \quad \delta_i^{(t)} = \hat{f}_i(x^{(t)}) + \gamma \hat{V}_i(x^{(t+1)}; \bar{\theta}_i) - \hat{V}_i(x^{(t)}; \theta_i). \tag{10}$$

Here $\hat{f}(x^{(t)})$ is in general an empirical estimate of the expected return $f(x^{(t)})$ based on a batch of Monte-Carlo rollouts, although in our experiments we use the exact expected return. The target involves $\bar{\theta}$, typically a target network that lags behind $\theta$ (Mnih et al., 2015).

As training progresses, (9) approaches the proposed update (7). The algorithm can be viewed as a consistent version of Meta-MAPG (Kim et al., 2021), and the value-learning counterpart to policy gradient-based M-FOS (Lu et al., 2022) with local meta-policy $\pi_i$.

### 3.3 Q-LEARNING INTERPRETATION

In this section we establish a theoretical link between the gradient $\nabla_{x_i} V_i$ and the action that greedily (if locally) maximizes the state-action value $Q_i$ given by

$$Q_i(x, x_i') = \mathbb{E}_{x_{-i}'} V_i(x_i', x_{-i}'), \quad \text{or equivalently} \quad Q_i(x, x_i + \Delta_i) = \mathbb{E}_{\Delta_{-i}} V_i(x + \Delta),$$

where we have defined $\Delta_i = x_i' - x_i$ so as to write the $Q$-function in terms of policy changes. Next, we construct a first-order Taylor approximation $\tilde{Q}_i$ of $Q_i$ around $x$:

$$\tilde{Q}_i(x, x_i + \Delta_i) = V_i(x) + \mathbb{E}_{\Delta_{-i}} \Delta^\top \nabla_x V_i(x) \tag{11}$$

$$= V_i(x) + \Delta_i^\top \nabla_{x_i} V_i(x) + \mathbb{E}_{\Delta_{-i}} \Delta_{-i}^\top \nabla_{x_{-i}} V_i(x). \tag{12}$$

**Algorithm 1** Basic Meta-Value Learning.

**Require:** Learning rates $\eta, \alpha$, rollout length $T$, fixed discount rate $\gamma$.
  Initialize models $\theta_i$ for all players $i$.
  **while** $\theta$ has not converged **do**
    Initialize policies $x^{(0)}$.
    **for** $t = 0, \ldots, T$ **do**
      $x^{(t+1)} = x^{(t)} + \alpha \bar{\nabla}_x \hat{V}(x^{(t)}; \theta)$
    **end for**
    **for** players $i \in \{1, \ldots, P\}$ **do**
      $\theta_i \leftarrow \theta_i - \eta \nabla_{\theta_i} \frac{1}{T} \sum_t (\delta_i^{(t)})^2$
    **end for**
  **end while**

**Algorithm 2** Basic Meta-Value Learning versus unknown meta-policies.

**Require:** Learning rates $\eta, \alpha$, rollout length $T$, fixed discount rate $\gamma$.
  Initialize model $\theta_i$ for player $i$.
  **while** $\theta_i$ has not converged **do**
    Initialize policies $x^{(0)}$.
    **for** $t = 0, \ldots, T$ **do**
      $x_i^{(t+1)} = x_i^{(t)} + \alpha \nabla_{x_i} \hat{V}_i(x^{(t)}; \theta_i)$
      $x_{-i}^{(t+1)} \sim \pi_{-i}(\,\cdot\mid x^{(t)})$
    **end for**
    $\theta_i \leftarrow \theta_i - \eta \nabla_{\theta_i} \frac{1}{T} \sum_t (\delta_i^{(t)})^2$
  **end while**

This approximation is not justified in general as there is no reason to expect $\Delta$ to be small, particularly the opponent updates $\Delta_{-i}$ which we do not control. It is however justified when all players are *learners*, i.e. they make local updates with small $\Delta$.

We now proceed to maximize (11) to find the argmax of $\tilde{Q}_i$. In doing so, we must include a locality constraint to bound the problem. If we use a soft norm penalty, we arrive at exactly our update:

$$\text{argmax}_{\Delta_i} \tilde{Q}_i(x, x_i + \Delta_i) - \tfrac{1}{2\alpha}\|\Delta_i\|^2 = \alpha \nabla_{x_i} V_i(x).$$

However, we may wish instead to apply a hard norm constraint, which would admit a treatment from the perspective of a game with a well-defined *local* action space. Either way, the argmax update will be proportional to $\nabla_{x_i} V_i(x)$, which lends an interpretation to our use of the gradient in (7).

In conclusion, our proposed update locally maximizes a local linearization $\tilde{Q}$ of the $Q$-function. Our method is thus related to independent $Q$-learning (Watkins & Dayan, 1992; Busoniu et al., 2008), which we must point out is not known to converge in general-sum games. It nevertheless does appear to converge reliably in practice, and we conjecture that applying it on the level of optimization effectively simplifies the interaction between the agents' learning processes.

## 4 PRACTICAL CONSIDERATIONS

We use a number of established general techniques to improve the dynamics of value function approximation (Hessel et al., 2018). The prediction targets in (10) are computed with a target network (Mnih et al., 2015) that is an exponential moving average of the parameters $\theta_i$. We use distributional reinforcement learning with quantile regression (Dabney et al., 2018). Instead of the fully-bootstrapped TD(0) error, we use $\lambda$-returns (Sutton & Barto, 2018) as the targets, computed individually for each quantile.

The rest of this section describes some additional techniques designed to mitigate the downsides of bootstrapping and to encourage generalization. Algorithm 3 in Appendix B lays out the complete learning process with these techniques included.

### 4.1 REFORMULATION AS A CORRECTION

The use of a model $\hat{V}$ introduces a bias, particularly early on in training when its gradients $\bar{\nabla}\hat{V}$ are meaningless. We provide a variant of the method that provides a correction to the original game $f$ rather than replacing it entirely. Instead of modeling

$$V(x) = f(x) + \gamma \, \mathbb{E}_{x'} V(x'),$$

we may model $U(x) = \mathbb{E}_{x'} V(x')$. We can derive a Bellman equation for $U$:

$$U(x) = \mathbb{E}_{x'} V(x') = \mathbb{E}_{x'} f(x') + \gamma \, \mathbb{E}_{x''} V(x'')$$

$$= \mathbb{E}_{x'} f(x') + \gamma U(x').$$

Now agents follow the gradient field $\nabla_{x_i} f_i(x) + \gamma \nabla_{x_i} \hat{U}_i(x)$, and we minimize

$$\sum_t (\delta_i^{(t)})^2 \quad \text{where} \quad \delta_i^{(t)} = \hat{f}_i(x^{(t+1)}) + \gamma \hat{U}_i(x^{(t+1)}; \bar{\theta}_i) - \hat{U}_i(x^{(t)}; \theta_i)$$

with respect to the parameters $\theta_i$ of our model $\hat{U}_i(x; \theta_i)$. This loss is the same as that for $\hat{V}$ but with a time shift on the $f$ term.

This variant is more strongly grounded in the game $f$, as the use of the original gradient $\nabla_{x_i} f_i(x)$ guards against poor initialization and spurious drift in the model. A drawback of this approach is that now the naive gradient term $\nabla_{x_i} f_i(x)$ will in general have to be estimated by REINFORCE (Williams, 1992). In all of our experiments we are able to compute $\nabla_{x_i} f_i$ exactly.

### 4.2 Variable Discount Rates

We set up the model (be it $\hat{U}$ or $\hat{V}$) to condition on discount rates $\gamma_i$, so that we can train it for different rates and even rates that differ between the players. This is helpful because it forces the model to better understand the given policies, in order to distinguish policies that would behave the same under some fixed discount rate but differently under another. During training, we draw $\gamma_i \sim \text{Beta}(1/2, 1/2)$ from the standard arcsine distribution to emphasize extreme values.

Varying $\gamma$ affects the scale of $U, V$ and hence the scale of our approximations to them. This in turn changes the effective learning rate when we take gradients. To account for this, we could normalize the outputs and gradients of $\hat{U}, \hat{V}$ by scaling by $1 - \gamma$ before use. However, we instead choose to multiply the meta-reward term $f(x)$ in the Bellman equations by $1 - \gamma$:

$$V_i(x; \gamma) = (1 - \gamma_i) f_i(x) + \gamma_i \mathbb{E}_{x'} V_i(x'; \gamma)$$

$$U_i(x; \gamma) = \mathbb{E}_{x'} (1 - \gamma_i) f_i(x') + \gamma_i V_i(x'; \gamma)$$

This ensures our models learn the normalized values instead, which fall in the same range as $f(x)$. Appendix A has a derivation.

### 4.3 Exploration

Our model $\hat{V}$ provides a deterministic (meta)policy for changing the inner policies $x$. Effective value learning, however, requires exploration as well as exploitation. A straightforward way to introduce exploration into the system is to perturb the greedy transition in (9) with some additive Gaussian noise (Heess et al., 2015). However, this leads to a random walk that fails to systematically explore the joint policy space. Instead of perturbing the (meta)actions, we perturb the (meta)policy by applying noise to the parameters $\theta_i$, and hold the perturbed policy fixed over the course of an entire exploration trajectory $\tilde{x}^{(0)}, \ldots, \tilde{x}^{(T)}$. Specifically, we randomly flip signs on the final hidden units of $\hat{V}$; this results in a perturbed value function that incentivizes different high-level characteristics of the inner policies $x$. The trajectories so collected are entirely off-policy and serve only to provide a diversity of states. In order to train our model on a given state $x$, we collect a short on-policy rollout with the unperturbed parameters $\theta$ and minimize TD error there.

## 5 Experiments

The method is evaluated on four environments. First, we demonstrate the advantage of looking farther ahead on a two-dimensional game that is easy to visualize. Next, we evaluate opponent shaping on the IPD, IMP and Chicken games by pitting MeVa head-to-head against Naive and LOLA agents, and M-MAML (a special exact case of Meta-MAPG due to Lu et al. (2022)).

### 5.1 Logistic Game

We analyze the behavior of several algorithms on the Logistic Game (Letcher, 2018), a two-player game where each player's policy is a single scalar value. Thus the entire joint policy space is a two-dimensional plane, which we can easily visualize. The game is given by the function[1]

$$f(x) = -\begin{pmatrix} 4\sigma(x_1)(1 - 2\sigma(x_2)) \\ 4\sigma(x_2)(1 - 2\sigma(x_1)) \end{pmatrix} - \frac{x_1^2 x_2^2 + (x_1 - x_2)^2 (x_1 + x_2)^2}{10000}. \tag{13}$$

---

[1]Letcher (2018) use the divisor 1000 in Eqn (13), however it does not match their plots.

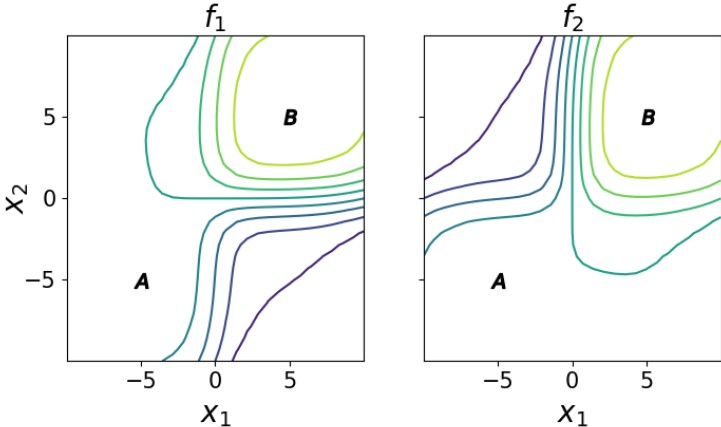

Figure 1: The Logistic Game. The left panel displays the contours of player 1's objective $f_1(x)$, the right panel similarly for player 2. Player 1's policy $x_1$ is a horizontal position, player 2's policy $x_2$ is a vertical position. Both players prefer solution $B$ over solution $A$, but cannot unilaterally go there. Naive learning converges to whichever solution is closest upon initialization.

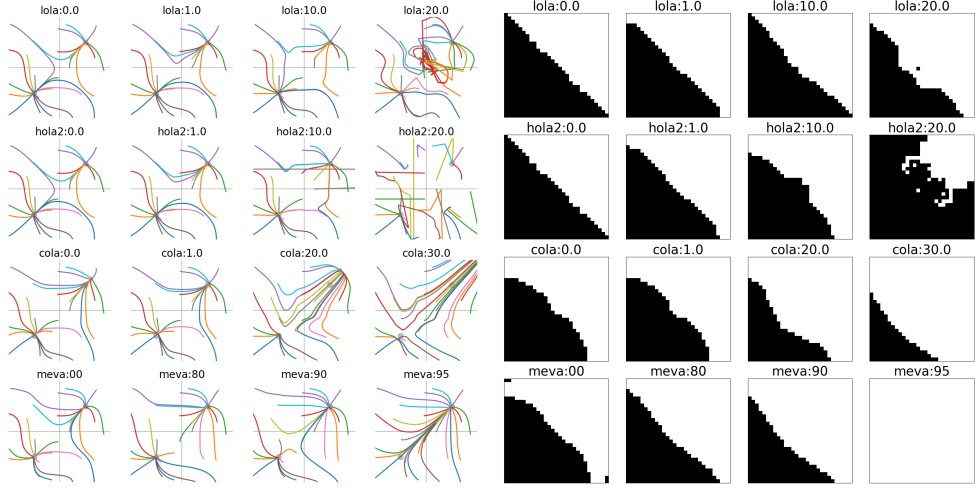

(a) Optimization trajectories. We took a random set of policy pairs and, for each panel, optimized them according to the algorithm under consideration. Each curve shows an optimization trajectory, typically finishing in or close to either $A$ or $B$.

(b) Basins of attraction. For each panel, we took a grid of policy space points and optimized them according to the algorithm under consideration. White cells indicate that the corresponding point ended up in the positive quadrant $x_1, x_2 > 0$, black cells ended up in other quadrants (typically the negative quadrant).

Figure 2: Logistic Game behaviors of different algorithms (rows) with different settings (columns). For LOLA/HOLA/COLA, the parameter is the learning rate $\alpha$. Increasing it leads to instability. For MeVa, the parameter is the meta-discount rate $\gamma$, which gradually smooths out the landscape until it leads towards $B$ from anywhere.

Figure 1 shows the structure of the game. There are two stable fixed points – one in the lower left ($A$) and one in the upper right ($B$). Both players prefer $B$ to $A$, however to get from $A$ to $B$ requires coordination: the horizontal player prefers left if the vertical player plays low and vice versa.

We look at this game in terms of basins of attraction, and how different algorithms affect them (Figure 2b). Following naive gradients (LOLA/HOLA2 with $\alpha = 0$), players converge to whichever solution is nearest; the basins of attraction meet at a diagonal line through the origin. LOLA grows

Table 1: Payoffs for the matrix games considered.

(a) Iterated Prisoner's Dilemma

|   | $A$ | $B$ |
|---|---|---|
| $A$ | $(-1,-1)$ | $(-3,\ 0)$ |
| $B$ | $(\ 0,-3)$ | $(-2,-2)$ |

(b) Iterated Matching Pennies

|   | $A$ | $B$ |
|---|---|---|
| $A$ | $(+1,-1)$ | $(-1,+1)$ |
| $B$ | $(-1,+1)$ | $(+1,-1)$ |

(c) Chicken Game

|   | $A$ | $B$ |
|---|---|---|
| $A$ | $(\ 0,\ 0)$ | $(\ -1,\ +1)$ |
| $B$ | $(+1,-1)$ | $(-100,-100)$ |

the basin of the preferred solution $B$, but only slightly and increasing the extrapolation step size $\alpha$ does not help much. HOLA2 grows the basin of $B$ around the edges, but suffers from instabilities around the origin (a saddlepoint). We found HOLA3 to be significantly worse than HOLA2 and did not pursue that direction further. COLA (our implementation) makes significant improvements around the edges and around the origin, but overshoots $B$ as $\alpha$ increases. Finally, MeVa is able to make the basin of $B$ arbitrarily large. When $\gamma > 0.9$, it converges to the preferred solution $B$ from anywhere in the surveyed area. We also show some actual optimization trajectories in Figure 2a.

Experiment details can be found in Appendix C.

## 5.2 MATRIX GAMES

We evaluate our method on several repeated matrix games by going head-to-head with naive learners, LOLA and M-MAML. M-MAML is a variant of MetaMAPG due to Lu et al. (2022) that uses exact gradients. We do not provide direct comparison with M-FOS as doing so requires training M-FOS and MeVa jointly; it is unclear how to do this fairly. Instead, we compare our behaviors versus Naive, LOLA and M-MAML with those of M-FOS.

We use the same setup as Lu et al. (2022): policies $x_i \in \mathbb{R}^5$ consist of five binary logits, corresponding to the probability of playing action $A$ or $B$ in each of five possible states (the initial state $\emptyset$ and the previous joint action $AA$, $AB$, $BA$, $BB$). We use the exact value function given by Foerster et al. (2018a) with discount rate 0.96 (not to be confused with our meta-discount rate $\gamma$).

The games differ only in their payoff matrices (Table 1), however IMP stands out as being not symmetric but antisymmetric. Lu et al. (2022) performed comparisons as though it were symmetric, that is, as though (meta)policies trained to (meta)play player 1 could be used to (meta)play player 2. Particularly, the numbers they report for M-MAML are meaningless. The numbers we report do not have this problem, except M-MAML vs M-FOS wich was measured using their code, and which we omit from the table for this reason. Please see Appendix D for more discussion of the issue.

On the **Iterated Prisoner's Dilemma** (Table 2a), MeVa extorts the naive learner (details in Appendix F), and LOLA to a small extent. The behavior is similar to that of M-FOS, although M-FOS leads the naive agent to accept returns below -2, indicating dynamical exploitation.

On the **Iterated Matching Pennies** game (Table 2b), MeVa exploits naive and LOLA learners, more so than M-FOS. ZD-extortion is not possible in zero-sum games, so MeVa exhibits dynamical exploitation.

On the **Chicken Game** (Table 2c), LOLA exploits every opponent except M-MAML, but does poorly against itself (as also observed by Lu et al. (2022)). M-MAML similarly exploits its opponents by taking on an aggresive initial policy. MeVa exploits the naive learner M-MAML while avoiding disasters against itself.

Overall, we find that MeVa is competitive with M-FOS on these games. Despite the restriction to local meta-actions, MeVa finds ZD-extortion and even dynamical exploitation on games that permit it. Further detail, including standard errors on these results, can be found in Appendix E.

Table 2: Head-to-head comparison of meta-policies on repeated matrix games. For each pairing we report the return of the row player, averaged across a batch of trials. Appendix E reports standard errors on these numbers.

(a) Iterated Prisoner's Dilemma. MeVa extorts the naive learner and MMAML, and is slightly extorted by LOLA.

|       | Naive | LOLA | MMAML | MFOS | MeVa |
|-------|-------|------|-------|------|------|
| Naive | -1.99 | -1.38 | -1.52 | -2.02 | -2.00 |
| LOLA  | -1.36 | -1.04 | -0.97 | -1.02 | **-1.03** |
| MMAML | -1.40 | -1.29 | -1.22 |       | -1.99 |
| MFOS  | **-0.56** | **-1.02** |       | -1.01 |       |
| MeVa  | **-0.55** | -1.15 | **-0.53** |       | -1.05 |

(b) Iterated Matching Pennies. MeVa is able to exploit both the naive learner and LOLA, but not MMAML. This exploitation must be dynamical in nature, as ZD-extortion cannot occur in zero-sum games.

|       | Naive | LOLA | MMAML | MFOS | MeVa |
|-------|-------|------|-------|------|------|
| Naive | 0.01  | 0.03 | -0.10 | -0.20 | -0.24 |
| LOLA  | -0.03 | 0.03 | -0.05 | -0.19 | -0.30 |
| MMAML | 0.10  | 0.05 | **-0.00** |       | **-0.01** |
| MFOS  | 0.20  | 0.19 |       | **0.00** |       |
| MeVa  | **0.24** | **0.30** | 0.01 |       | **-0.00** |

(c) Chicken Game. MeVa exploits the naive learner and MMAML, while avoiding disasters against itself. LOLA exploits every opponent except MMAML, but does poorly against itself.

|       | Naive | LOLA | MMAML | MFOS | MeVa |
|-------|-------|------|-------|------|------|
| Naive | -0.05 | -0.40 | -0.99 | -1.03 | -0.98 |
| LOLA  | 0.38  | -1.64 | -0.80 | **0.79** | **0.14** |
| MMAML | **0.98** | **0.78** | -0.45 |       | -0.91 |
| MFOS  | 0.97  | -1.16 |       | -0.01 |       |
| MeVa  | 0.96  | -0.23 | **0.17** |       | -0.08 |

## 6  LIMITATIONS

The meta-value function is a scalar function over (joint) policies. In practice, policies will often take the form of neural networks, and so will our meta-value function approximation. Conditioning neural nets on other neural nets is a major challenge (Harb et al., 2020). In addition, the large parameter vectors associated with neural nets quickly prohibit handling batched optimization trajectories.

During training and opponent shaping, we assume opponent parameters to be visible to our agent. This is not necessarily unrealistic – we learn and use the meta-value only as a means to the end of finding good policies $x$ for the game $f$ that can then be deployed in the wild without further training. Nevertheless, the algorithm could be extended to work with opponent models, or more directly, the model could observe policy behaviors instead of parameters.

The meta-discount rate $\gamma$, like LOLA's step size $\alpha$, is hard to interpret. Its meaning changes significantly with the learning rate $\alpha$ and the parameterization of both the model $\hat{V}$ and the policies $x$. Future work could explore the use of proximal updates, like POLA (Zhao et al., 2022) did for LOLA. Finally, it is well known that LOLA fails to preserve the Nash equilibria of the original game $f$. The method presented here shares this property.

## 7  CONCLUSION

We have introduced Meta-Value Learning (MeVa), a naturally consistent and far-sighted approach to learning with learning awareness. MeVa derives from a meta-game similar to that considered in prior work (Al-Shedivat et al., 2018; Kim et al., 2021; Lu et al., 2022), and can be seen as the $Q$-learning complement to the policy gradient of M-FOS (Lu et al., 2022), although we essentially choose different meta-action spaces. MeVa avoids explicitly modeling the continuous action space by parameterizing the action-value function implicitly through a state-value function.

MeVa's opponent-shaping capabilities are similar to those of M-FOS (Lu et al., 2022). Like M-FOS, we find ZD-extortion on the general-sum IPD, and dynamical exploitation on the zero-sum IMP.

The main weakness of the method as it stands is scalability, particularly to policies that take the form of neural nets. We aim to address this in future work using policy fingerprinting (Harb et al., 2020).

Finally, we note that although we develop our method in the context of multi-agent learning, it is a general meta-learning approach that readily applies to optimization problems with a single objective.

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

## A  NORMALIZED BELLMAN EQUATION

We introduce a slight tweak to the usual Bellman equation

$$V(s) = r + \gamma V(s')$$

to address a scaling issue. Suppose the rewards fall in the range $(0, 1)$. Then $V$ will take values in the range $(0, \frac{1}{1-\gamma})$. We can multiply through by $1 - \gamma$ to obtain a new equation for *normalized* values $\mathring{V}$ that fall in the same range as the rewards:

$$(1 - \gamma)V(s) = (1 - \gamma)r + \gamma(1 - \gamma)V(s')$$
$$\mathring{V}(s) = (1 - \gamma)r + \gamma\mathring{V}(s').$$

We use the normalized values primarily because we train a single model for all values of $\gamma$, so we require the scale of the outputs of our model to be invariant to it. Moreover, we use gradients $\bar{\nabla}\mathring{V}$ whose scale would otherwise be affected by a change in $\gamma$, which in turn would change the effective learning rate. We also found it helpful when using distributional RL with a fixed binning – the convex-combination form of the right-hand side guarantees it will fall within the same range as the left-hand side. Finally, we find the normalized values easier to interpret.

## B  DETAILED ALGORITHM DESCRIPTION

Algorithm 3 describes the learning algorithm in detail, including the modifications discussed in §4. The algorithm is written from the perspective of a single MeVa learner $i$ learning a single model $\theta_i$ that predicts the meta-value of *all* players, regardless of whether they are meta-value learners. These auxiliary predictions are cheap to include, and they likely improve sample complexity. When all learners are meta-value learners, we share these models across players.

After introducing quantile distributional RL, we have a model $\tilde{U}_i(x; \gamma, \theta_i) \in \mathbb{R}^M$ that produces a vector of $M$ quantiles. We denote by $\hat{U}_i(x; \gamma, \theta_i)$ the expectation over the quantile distribution,

$$\hat{U}_i(x; \gamma, \theta_i) = \frac{1}{M} \sum_{m=1}^{M} \tilde{U}_i(x; \gamma, \theta_i)_m.$$

The divergence $D(\hat{y}, y)$ is the quantile regression loss from Dabney et al. (2018) between the predicted quantiles $\hat{y}$ and the target quantiles $y$.

## C  LOGISTIC GAME DETAILS

On the Logistic Game, we use the $\hat{V}$ formulation because the game is simple enough. While conceptually, each agent maintains their own model $\hat{V}_i(x; \gamma, \theta_i)$, the implementation combines the computation of both models. The structure of the resulting single model can be seen in the following diagram:

We first feed the $x_i, \gamma_i$ pairs into a multi-layer perceptron (MLP) to obtain a representation $z_i$ of each agent. Then for each agent we concatenate their own representation with that of their opponent. This is run through a second MLP which outputs the quantile estimates.

The dotted lines in the diagram indicate parameter sharing between the two players ($\theta_1 = \theta_2$), which exploits the symmetry of the games under consideration (specifically $f_1(x_1, x_2) = f_2(x_2, x_1)$) to improve sample efficiency of the learning process.

---

**Algorithm 3** Meta-Value Learning versus unknown meta-policies, incorporating the techniques discussed in §4.

---

**Require:** Learning rates $\eta, \alpha$, episode length $T$, stride $k$, bootstrapping rate $\lambda$, target net inertia $\rho$.
Initialize model parameters $\theta_i$ and target network $\bar{\theta}_i \leftarrow \theta_i$.
  **while** $\theta_i$ has not converged **do**                                                              ▷ outer loop
      Initialize policies $\tilde{x}^{(0)}$; draw $\tilde{\gamma}$ (§4.2); draw $\tilde{\theta}$ (§4.3).
      **for** $t = 0, \ldots, T$ **do**                                           ▷ inner loop

$$\tilde{x}_i^{(t+1)} = \tilde{x}_i^{(t)} + \alpha \nabla_{x_i} \left( (1 - \tilde{\gamma}_i) \hat{f}_i(\tilde{x}^{(t)}) + \tilde{\gamma}_i \hat{U}_i(\tilde{x}^{(t)}; \tilde{\gamma}, \tilde{\theta}_i) \right)$$

$$\tilde{x}_{-i}^{(t+1)} \sim \pi_{-i}( \cdot \mid \tilde{x}^{(t)})$$

         **if** $t$ is divisible by $k$ **then**
            Let $x^{(0)} = \tilde{x}^{(t)}$; draw $\gamma$ (§4.2).                 ▷ prepare to train $\hat{U}$ on $\tilde{x}^{(t)}$
            **for** $\tau = 0 \ldots k - 1$ **do**            ▷ produce an on-policy trajectory of length $k$

$$x_i^{(\tau+1)} = x_i^{(\tau)} + \alpha \nabla_{x_i} \left( (1 - \gamma_i) \hat{f}_i(x^{(\tau)}) + \gamma_i \hat{U}_i(x^{(\tau)}; \gamma, \theta_i) \right)$$

$$x_{-i}^{(\tau+1)} \sim \pi_{-i}( \cdot \mid x^{(\tau)})$$

            **end for**
            **for** players $j \in \{1, \ldots, P\}$ **do**       ▷ compute $\lambda$-return distributions for all players

$$Y_j^{(k)} = \tilde{U}_j(x^{(k)}; \gamma, \bar{\theta}_i) \in \mathbb{R}^M$$

               **for** $\tau = k - 1 \ldots 0$ **do**

$$Y_j^{(\tau)} = (1 - \gamma_j) \hat{f}_j(x^{(\tau+1)}) + \gamma_j \left( (1 - \lambda) \tilde{U}_j(x^{(\tau+1)}; \gamma, \bar{\theta}_i) + \lambda Y_j^{(\tau+1)} \right)$$

               **end for**
            **end for**
            $\theta_i \leftarrow \theta_i - \eta \nabla_{\theta_i} \sum_{j=1}^{P} \mathcal{L}_j(\theta_i)$ where $\mathcal{L}_j(\theta_i) = \frac{1}{k} \sum_{\tau=0}^{k-1} D(\tilde{U}_j(x^{(\tau)}; \gamma, \theta_i), Y_j^{(\tau)})$.
            $\bar{\theta}_i \leftarrow \bar{\theta}_i + (1 - \rho)(\theta_i - \bar{\theta}_i)$                     ▷ update target network
         **end if**
      **end for**
  **end while**

---

The MLPs consist of a residual block (Srivastava et al., 2015; He et al., 2016) sandwiched between two layer-normalized GELU (Hendrycks & Gimpel, 2016) layers. The residual block uses a layer-normalized GELU as nonlinearity, and uses learned unitwise gates to merge with the linear path. Each layer has 64 units.

We use the same model structure for COLA, albeit without quantile regression. In the case of COLA we pass the inner learning rate $\alpha$ in place of $\gamma$, and we trained a single model for values of $\alpha \sim U[0, 10]$.

Policies $\tilde{x}_i^{(0)} \sim U(-8, +8)$ are initialized uniformly on an area around the origin.

We used hyperparameters $\alpha = 1, M = 16, T = k = 50, \lambda = 0.9$. In this experiment we do not use a target network, nor do we use exploration (i.e. $\tilde{\theta} = \theta$). The model is trained for 5000 outer loops using Adam (Kingma & Ba, 2014) with learning rate $\eta = 10^{-3}$ and batch size 128. Training takes about five minutes on a single GPU.

## D   SYMMETRY OF THE MATRIX GAMES

The Iterated Prisoner's Dilemma (IPD) and the Chicken Game are *symmetric* games, which means that permuting the players permutes the returns: $f_1(x_1, x_2) = f_2(x_1, x_2)$.[2] This is a consequence of the players' payoff matrices being related by a transposition. The Iterated Matching Pennies game (IMP), however, is *antisymmetric*, meaning that permutation introduces a sign switch as well: $f_1(x_1, x_2) = -f_2(x_1, x_2)$.

The implication of this is that while on IPD and the Chicken Game, policies trained to play one side can be *transferred* to play the other side just as well, the same is *not* true for IMP. In fact the *opposite* is true for IMP: transferred policies are as bad as the original policies were good! A corollary is that meta-policies trained to meta-play one side of the IMP meta-game can *not* be transferred to meta-play the other. This can lead to bugs in algorithmic comparisons when meta-policies are trained upfront and stored, and later called up to meta-play either side. We noticed this issue in our own experiments, and see it in the experiments of Lu et al. (2022) as well, where they train M-MAML to play as player 1 and later evaluate it as player 2.

Simple meta-policies like Naive and LOLA have no parameters to be learned, and training and evaluation of the inner policies $x$ is usually done in a single procedure, with no opportunity for inadvertent transfer. M-FOS, M-MAML and MeVA, however, are somewhat expensive to train and as such it is better to train them once upfront and then perform evaluations and other analyses later. When a tournament involves only one parametric meta-policy, it can be trained to play one side exclusively and be evaluated accordingly, and the other side can be played by the nonparametric meta-opponents. However, when a tournament involves two (or more) parametric meta-policies, at least one of them must be trained twice (so as to be able to play both sides).

Lu et al. (2022) compared M-FOS and M-MAML, but neither was trained to play both sides (except in M-FOS vs M-FOS), and M-MAML was incorrectly transferred to play both sides. As such, their results involving M-MAML are meaningless. In our experiments, we compare MeVa and M-MAML, and trained two sets of M-MAML meta-policies. This allows MeVa to play player 1 (except in MeVa vs MeVa), and M-MAML to play either side with different models.

## E   TOURNAMENT DETAILS

Table 3 reports tournament results with standard errors. For each game and opposing meta-policy (naive or LOLA or MeVa, with M-MAML counting as naive), we train 10 models from different seeds. We use meta-discount rate $\gamma = 0.95$, and learning rates $\alpha = 25$ (except on the Chicken Game, where we use $\alpha = 1$ to accommodate the large payoffs).[3] For M-FOS we found different numbers than those reported by Lu et al. (2022), so to be sure we trained 30 models with 4096 policy

---

[2]Making this hold exactly requires modifying the game so that player 2 always sees the state as if it were player 1; a common practice in prior work.

[3]Unlike Foerster et al. (2018a); Lu et al. (2022), we work with the *normalized* value, and hence our learning rates must be $1/(1 - 0.96) = 25$ times larger to match.

Table 3: Head-to-head comparison of learning rules on repeated matrix games. For each pairing, we report the average return ($\pm$ standard error) of the row player. Numbers not involving M-FOS are the result of playing 1024 policy pairs. In the case of MeVa and M-MAML these are spread over 10 models trained from different seeds. For M-FOS, we trained 30 models and played 4096 matches per model; here the standard error is only across the models as the matches were already averaged. We have omitted comparisons between M-FOS and M-MAML due to the asymmetry bug described in Appendix D, and between M-FOS and MeVa due to methodological complications.

(a) Iterated Prisoner's Dilemma.

|        | Naive         | LOLA          | MMAML         | MFOS          | MeVa          |
|--------|---------------|---------------|---------------|---------------|---------------|
| Naive  | -1.99$\pm$0.00 | -1.38$\pm$0.01 | -1.52$\pm$0.02 | -2.02$\pm$0.06 | -2.00$\pm$0.00 |
| LOLA   | -1.36$\pm$0.01 | -1.04$\pm$0.00 | -0.97$\pm$0.01 | -1.02$\pm$0.00 | **-1.03**$\pm$0.00 |
| MMAML  | -1.40$\pm$0.01 | -1.29$\pm$0.01 | -1.22$\pm$0.01 |               | -1.99$\pm$0.00 |
| MFOS   | **-0.56**$\pm$0.03 | **-1.02**$\pm$0.00 |               | **-1.01**$\pm$0.00 |               |
| MeVa   | **-0.55**$\pm$0.00 | -1.15$\pm$0.00 | **-0.53**$\pm$0.00 |               | -1.05$\pm$0.00 |

(b) Iterated Matching Pennies.

|        | Naive         | LOLA          | MMAML         | MFOS          | MeVa          |
|--------|---------------|---------------|---------------|---------------|---------------|
| Naive  | 0.01$\pm$0.01 | 0.03$\pm$0.02 | -0.10$\pm$0.01 | -0.20$\pm$0.00 | -0.24$\pm$0.01 |
| LOLA   | -0.03$\pm$0.02 | 0.03$\pm$0.02 | -0.05$\pm$0.01 | -0.19$\pm$0.00 | -0.30$\pm$0.01 |
| MMAML  | 0.10$\pm$0.01 | 0.05$\pm$0.01 | **-0.00**$\pm$0.01 |               | **-0.01**$\pm$0.00 |
| MFOS   | 0.20$\pm$0.00 | 0.19$\pm$0.00 |               | **0.00**$\pm$0.01 |               |
| MeVa   | **0.24**$\pm$0.01 | **0.30**$\pm$0.01 | **0.01**$\pm$0.00 |               | **-0.00**$\pm$0.00 |

(c) Chicken Game.

|        | Naive         | LOLA          | MMAML         | MFOS          | MeVa          |
|--------|---------------|---------------|---------------|---------------|---------------|
| Naive  | -0.05$\pm$0.02 | -0.40$\pm$0.02 | -0.99$\pm$0.00 | -1.03$\pm$0.00 | -0.98$\pm$0.00 |
| LOLA   | 0.38$\pm$0.02 | -1.64$\pm$0.37 | -0.80$\pm$0.02 | **0.79**$\pm$0.12 | **0.14**$\pm$0.02 |
| MMAML  | **0.98**$\pm$0.00 | **0.78**$\pm$0.02 | -0.45$\pm$0.04 |               | -0.91$\pm$0.04 |
| MFOS   | 0.97$\pm$0.00 | -1.16$\pm$0.12 |               | -0.01$\pm$0.13 |               |
| MeVa   | 0.96$\pm$0.00 | -0.23$\pm$0.02 | **0.17**$\pm$0.04 |               | -0.08$\pm$0.02 |

pairs each, using their code and hyperparameters ($\alpha = 25$ on IPD and the Chicken Game, $\alpha = 2.5$ on IMP). We also trained 10 seeds of M-MAML (our implementation) using $\alpha = 25$ on IPD and Chicken and $\alpha = 2.5$ on IMP. Each paired comparison not involving M-FOS is the result of training 1024 policy pairs (spread across model seeds) for 300 steps. Note that the comparison of M-FOS vs M-MAML is broken in the way described in §D.

As matrix games are a step up in complexity, we use the $\hat{U}$ formulation. We use a similar shared-model structure as for the Logistic Game (see the diagram in the previous section), but introduce an elementwise scale and shift on $z_1, z_2$ that is *not* shared, in order to break the equivariance when the opponent is not a MeVa agent. This makes it straightforward to still model the opponent's own meta-value, which as an auxiliary task may help learn the primary task. We also break the equivariance for the IMP game, which is antisymmetric.

In the final nonlinear layer, the GELU is replaced by a hyperbolic tangent – a signed nonlinearity that enables our exploration scheme. During exploration, we flip signs on the units in this layer with Bernoulli probability $1/16$ (see §4.3).

Policy logits are initialized from a standard Normal distribution, i.e. $\tilde{x}^{(0)} \sim \mathcal{N}(0, I)$.

We used hyperparameters $M = 64, T = 100, k = 10, \rho = 0.99, \lambda = 0.9$. The model is trained for 1000 outer loops, using batch size 128 and AdamW (Loshchilov & Hutter, 2018) with learning rate $10^{-3}$ and $10^{-2}$ weight decay. Training the model on a matrix game takes about five minutes on a single GPU.

For M-MAML we used Adam with learning rate $10^{-1}$, batch size 64 and (meta)rollouts of length 300.

## F  ZD-EXTORTION ON THE IPD

Figure 3 shows how MeVa shapes a naive opponent. MeVa immediately takes on a ZD-extortion policy to induce cooperation in the naive agent, indicated by a $> 1$ slope on the blue front. Then, it appears to dynamically exploit the naive learner to reach and maintain a position of maximum extortion (nearly vertical front).

## G  ABLATION

To evaluate the impact of the practical techniques from § 4, we perform an ablation experiment on the Iterated Prisoner's Dilemma. In Figure 4, we show the effect of each of the following modifications:

- Disabling the sign-flipping exploration scheme (§4.3), falling back to Noisy Networks (Fortunato et al., 2017) instead.
- Using $\lambda = 1$, the least amount of bootstrapping. In this case for each $x^{(\tau)}$ in the $k$-length onpolicy rollout, we minimize a $(k - \tau)$-step TD error (the maximal length available).
- Using $\lambda = 0$. Here only 1-step TD errors are used, and we additionally set the stride $k = 1$ to 1 to update more frequently. The total number of (meta)environment interactions is preserved.
- Disabling target networks, using the current model to compute the TD target (without differentiating through it).
- Disabling distributional RL in favor of emitting a point estimate trained with a Huber loss.
- Using the $V$ formulation instead of the $U$ formulation.
- Training for a fixed $\gamma = 0.95$.

For each configuration we train 10 models for 500 outer loops. We simultaneously train policies (a batch of 128 pairs) according to the model, and monitor their returns versus each other (as a measure of cooperation) as well as versus their best responses (as a measure of exploitability). The configurations vary in the extent to which they reduce the true error (estimated by the long-term TD error), while minimizing the loss (measured by the short-term TD error). The difference between

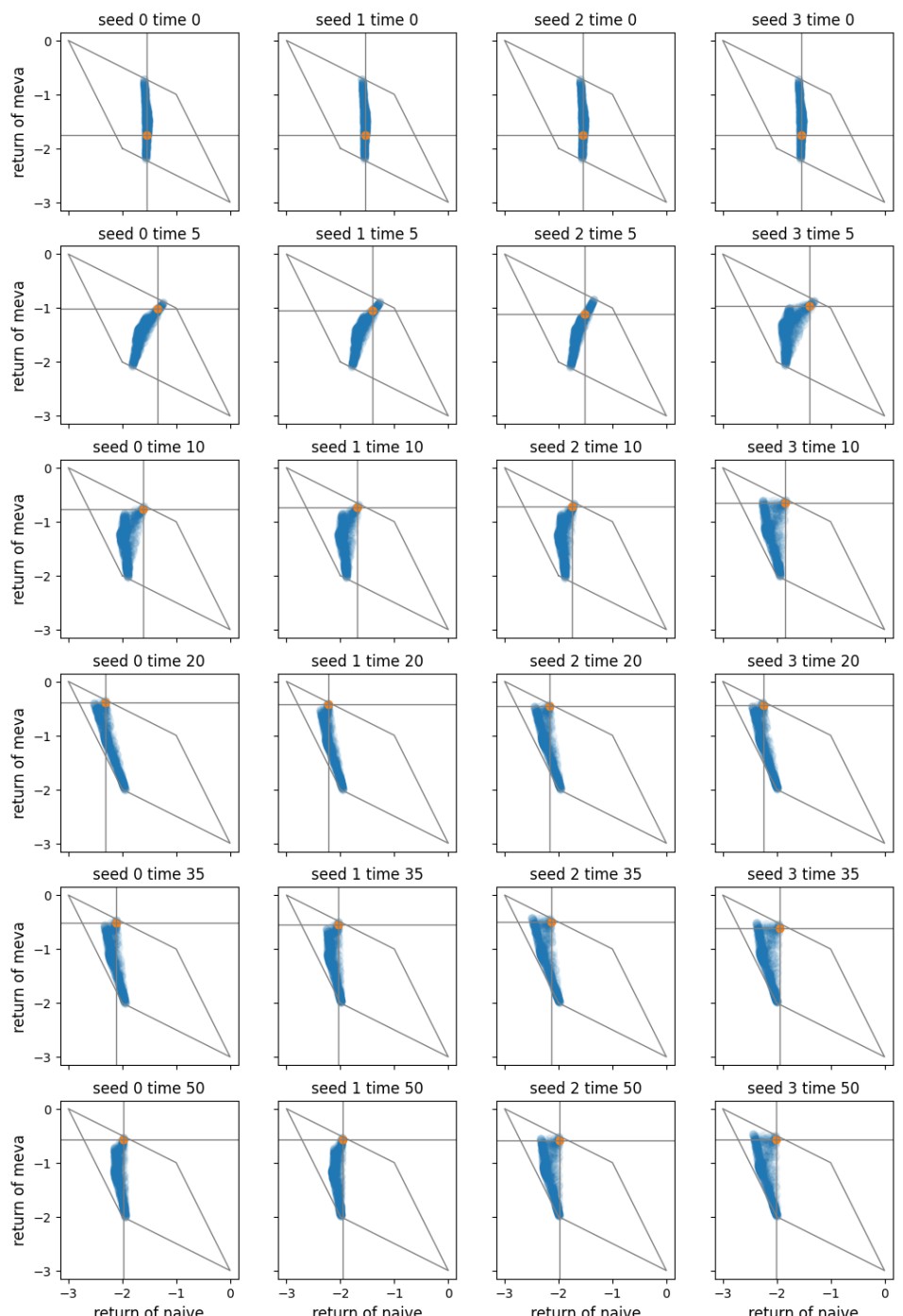

Figure 3: Meta-value agents extorting naive agents on the Iterated Prisoner's Dilemma. We train $\hat{U}$ from four different initializations (columns). Then, we initialize a pair of policies and show their behavior as they learn (rows). Each panel shows the polytope of possible game returns (gray outline), the subset of possible game returns given the current MeVa policy $x_1$ (blue scatterplot of return pairs obtained by pitting random policies against $x_1$), and the actual return pair given $x_1$ and $x_2$ (orange, with lines to help tell the angle of the rightmost front of the blue region).

these two is due to bootstrapping. Moreover, there are differences in the convergence to and stability of cooperation, as well as the exploitability of the policies.

Several of the modifications considered either breaks training, struggles to find and maintain cooperation or becomes exploitable. Exploration using random sign flips and increasing $\lambda$ all the way to 1 (minimal bootstrapping) have the least marginal impact. The use of $\lambda = 0$ runs the risk of learning short-term dynamics, and indeed it seems to do so. Bypassing target networks leads to drift during training. Disabling distributional RL hurts nonexploitability; it is unclear why this would be the case. The $V$ formulation struggles to get off the ground. Disabling variable-$\gamma$ training leads to mutual defection.

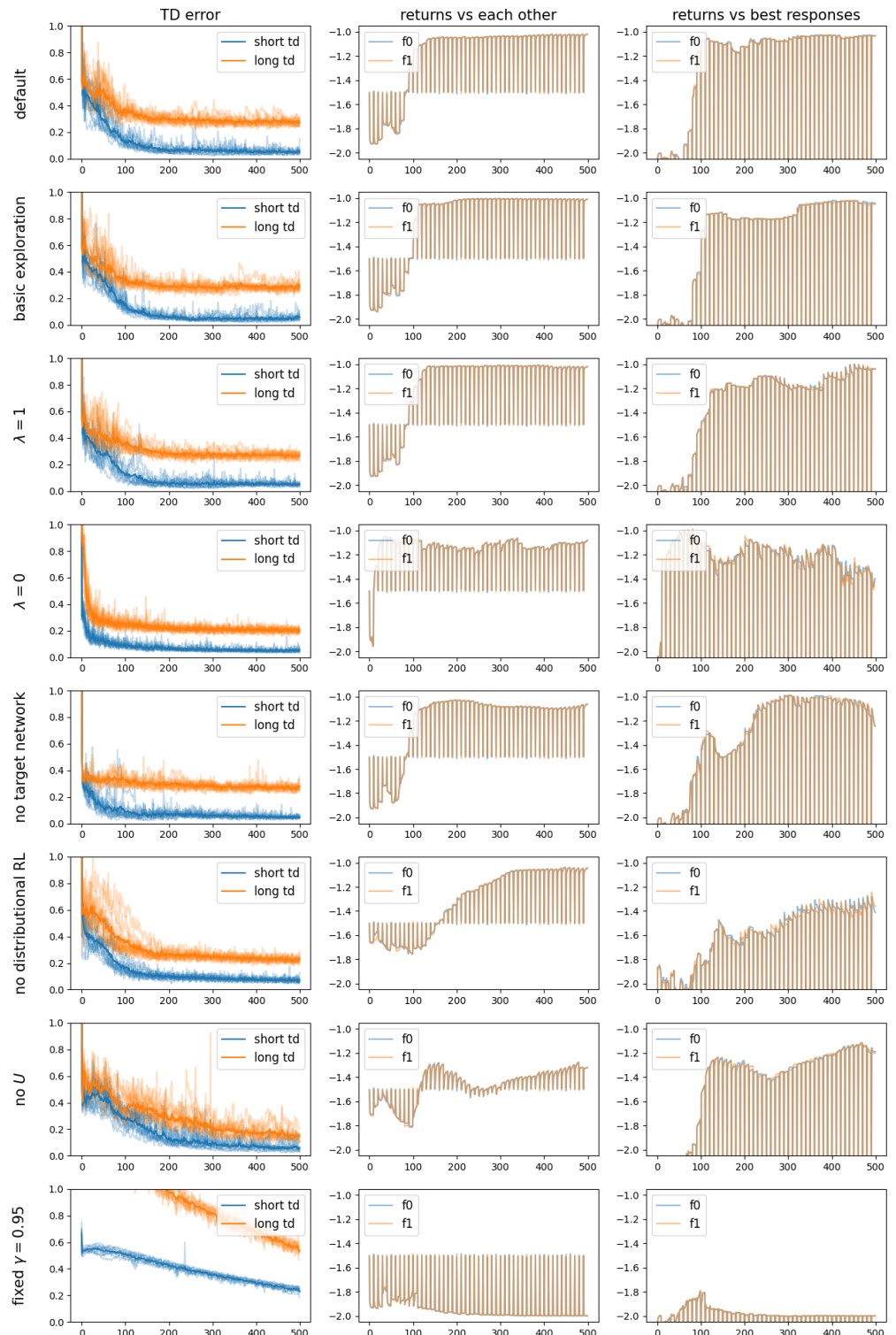

Figure 4: Ablation experiment on the Iterated Prisoner's Dilemma. We show the effect of using basic exploration instead of random sign flips, using minimal bootstrapping ($\lambda = 1$), using maximal bootstrapping ($\lambda = 0$), disabling distributional RL, disabling target networks, using the $V$ formulation over the $U$ formulation, and training with a fixed $\gamma = 0.95$. For each configuration (row) we train 10 models for 500 outer loops. In the leftmost column, we show short-term TD error (over $k = 10$ steps, as in training) and long-term TD error (over 100 steps, as a validation); the difference between these is due to bootstrapping. The horizontal axis measures number of outer loops performed. In the middle column, the returns $f(x)$ of agents that are being trained on the model (with $\gamma = 0.95$) and are reset every 10 outer loops. For those agents we continually test their exploitability; the third column shows their returns against agents trained to exploit them.

