# OpenReview forum: "Meta-Value Learning: a General Framework for Learning with Learning Awareness"
_ICLR.cc/2024/Conference — Submitted to ICLR 2024_

### Official Review · Reviewer_1uYc · 2023-10-30

**Soundness:** 3 good
**Presentation:** 3 good
**Contribution:** 2 fair
**Rating:** 5
**Confidence:** 4

**Summary:**

Recent works in multi-agent learning focus on reasoning about inter-agent interactions. One such algorithm is Learning with Opponent Learning Awareness (LOLA), which looks ahead utilizing gradient descent. The work builds on the LOLA framework by casting the problem as a meta-game of optimization. Meta-Value Learning (MeVa) learns a meta-value function (expected discounted return over future iterates) and applies a form of Q-learning to evade representing the continuous action space. The inner loop corresponds to collecting a policy optimization trajectory using gradient descent on iterates of the meta-value. The outer loop corresponds to learning the meta-value function by minimizing the TD error. MeVa incorporates design considerations such as empirical corrections in Bellman updates, variable discounting and gaussian noise as exploration. Results in a logistic game and repeated matrix games demonstrate improvements over LOLA agents.

**Strengths:**

* The paper is well-written and positioned within the learning with learning awareness literature.
* Authors have highlighted relevant design considerations and their motivation.

**Weaknesses:**

* **Variance in Iterates:** My main concern is the variance in iterates of the optimization process. Using a correction-based formulation leads MeVa to estimate $\bar{\nabla} f(x)$ using REINFORCE which is known to have high variance in the policy gradient. Specifically, it is essential that the gradient of the iterates is consistent. It would be helpful if the authors can highlight any practical considerations to mitigate high variance in the gradients. Authors could also provide standard deviations of different trials for Tables 2 and 3 in order to evaluate the effect of noisy gradients (if any).
* **Ablations:** Appendix F provides a range of ablations for design considerations utilized in MeVa. However, an important comparison would be to assess the importance of the meta-value function formulation itself. Authors should compare the surrogate formulation of Eq. 6 with the naive infinitely discounted sum of Eq. 5. Similarly, the choice of a sophisticated exploration strategy, which is Gaussian noise, could be evaluated by comparing it with standard exploration schedules used for TD learning such as $\epsilon$-greedy exploration or action noise. Current ablations only enable/disable exploration which do not reason about the choice of schemes and their importance.
* **Related Work:** The paper cites relevant works from multi-agent learning and opponent awareness literature. However, their organization within the paper could be improved. Authors could organize recent literature in a dedicated related works section or discuss the improvements of MeVa over prior methods (as done for LOLA in Introduction).

[1]. Foerster et al., "Learning with Opponent-Learning Awareness", AAMAS 2018.

**Questions:**

* How can the high variance of policy gradients in REINFORCE be tackled? Can you please discuss some practical considerations or the impact of noise on meta-value iterates? Can you please provide standard deviations for head-to-head comparisons in Tables 2 and 3?
* Can you please compare between meta-value function formulations of Eq. 5 and Eq. 6? What is the need for a sophisticated exploration strategy such as Gaussian noise when compared to standard exploration schedules such as $\epsilon$-greedy or action noise?
* Can you please organize the discussion on relevant works in a related works section? Alternatively, can the discussion be moved to a common section discussing the improvements/differences between MeVa and LOLA, HOLA, COLA and other opponent aware algorithms?

---

> ### Author Response · Authors · 2023-11-14
>
> Thank you for taking the time to review our paper. Please see our responses below:
> - Variance in iterates: as also noted in our response to reviewer c4Mz, MeVa does not rely on REINFORCE in principle. We did not use any REINFORCE gradients for the experiments because these games can be differentiated exactly. Thus the results in Tables 2 and 3 are not subject to REINFORCE variance.
>   However, we did note that our $U$ formulation requires a way of estimating $\nabla f$, and REINFORCE is the obvious choice. REINFORCE is in no way required though, as one could also use a model $\hat{f}$ to estimate $f$ and approximate $\nabla f$ by $\nabla \hat{f}$ (this is what the $V$ formulation with $\gamma=0$ would provide).
> - Ablations: unfortunately Eq 5 can not be used directly, as the rollouts $x_{>t}$ need to come from the meta-policy itself. Eq 5 (and 6 for that matter) are implicit, and can only be made explicit by fitting a model. We will however include a comparison (on the matrix games) to MetaMAPG, which can be seen as using Eq 5 but evaluating the expectation using naive learning as the meta-policy for the rollouts. For a fair comparison we will modify MetaMAPG to use exact gradients rather than policy gradients, which is the M-MAML variant discussed in the M-FOS paper.
> - Exploration: our exploration is done on the meta level, where actions are continuous, and it is not obvious what form the "default" exploration should take. The typical $\epsilon$-greedy approach is undefined, as there is no uniform distribution on $\mathbb{R}^n$. We could use the uniform distribution on the sphere to get uniform random directions, but then it is unclear how to scale these. We could add $\mathcal{N}(0,\epsilon^2)$ noise to $\bar{\nabla}\hat{V}$, however this scheme seems problematic in general as different parameters can have wildly different sensitivities (think neural networks), in which case we do not expect perturbing them with the same variance will lead to meaningful diversity. Nevertheless, we will rerun the ablation with this additive gaussian noise as the control.
> - We can reorganize related work, but please hear us out on the current structure:
>   - We discuss LOLA front and center because it is the seminal work in this area.
>   - In Section 1 we mention MetaMAPG and M-FOS because they are meta-level approaches like MeVa.
>   - In Section 2.2 we go over the idea of looking ahead, the *inconsistency* in LOLA and how it may be resolved. Here HOLA and COLA are briefly relevant.
>   - In Section 2.3 we discuss *looking further ahead*, which brings in the meta-game. Here Meta-(MA)PG and M-FOS become relevant.
>   - In Section 3.1 we introduce MeVa, and relate it to MetaMAPG and M-FOS.
>
>   We propose to make two improvements:
>   - We will add discussion of LOLA and COLA in Section 3.1.
>   - We will discuss COLA at the first mention of consistency ("It is self-consistent: ..."), which gives us an opportunity to clarify what is meant by (in)consistency.
>
> Thank you.

---

> > ### Comment · Reviewer_1uYc · 2023-11-22
> > **Response To Authors**
> >
> > I thank the authors for responding to my comments and refining the paper with connections to Q learning and M-MAML comparisons. After going through other reviews and response from authors, my following concern remains unadressed-
> > * **Ablations:** Authors provided ablations for exploration utilizing noisy networks. However, my concern was mostly centered towards the effect of the exploration strategy. Meta-learned agents are sensitive towards hyperparameters and design choices. Considering games where exact solutions can be learned, minimal exploration should be sufficient for players to reason about opponent actions. It would be helpful if authors could compare between random sign flips/noisy networks against minimal exploration schedules for future work. This would answer the question _To what degree does MeVa require exploration for finding exact solutions?_ Note that this does not undermine the efficacy of noisy networks as the paper identifies them as a useful strategy for meta games.
> >
> > On a general note, the overall quality of the paper is improved and the work presents useful insights. In future, authors could highlight the portions they have modified (with a different color) in the paper for brevity. Future work could consider additional matrix games (rock-paper-scissor and battle of the sexes) which have mixed strategy Nash Equilibria or games where $\epsilon$-Nash deviates from the true Nash Equilibrium ([[(100, 100), (0, 0)], [(1, 2), (1, 1)]]). Currently, only Matching Pennies presents a mixed strategy equilibrium. In my opinion, these experiments would help us understand the convergence and generality of MeVa. Considering symmetric games does enable transfer between agents which players in antisymmetric games may not enjoy. This does not hamper the utility of meta-learned agents. On the other hand, this is an important finding which could be further studied from an algorthmic perspective.

---

> > > ### Author Response · Authors · 2023-11-22
> > >
> > > Thank you for your response. First off -- you are right, we could have highlighted the changes. Apologies! Thank you for taking the time to read the paper again.
> > >
> > > We regret having misunderstood your request regarding minimal exploration. We believe minimal exploration would be sufficient *in principle*, but the practical implication of lowering the noise would be vastly more training time required to discover the eventual solution. For these matrix games it would not be so bad, but in more complex games with large and curved neural network parameter spaces, minimal local perturbations are going to struggle to meaningfully explore. It occurs to us that in our original ablation, we used no exploration at all as a baseline, and it did not particularly harm learning. This should answer your question of *to what degree MeVa requires exploration*: to no degree at all. That said, it would be premature to conclude this from matrix games where the parameter space is so simple.
> > >
> > > We do note that in Value-Gradient Learning [1], Fairbank showed that enforcing a Bellman equation on the gradient $\nabla V(x)$ of the value function relieves the need for local exploration. This would be an interesting avenue for future research, although optimizing a Bellman equation on gradients requires second-order gradients, which are expensive in terms of time, space and data. Our own interests for future work are in the direction of using a model like MeVa to understand and generalize across joint policy space, using its understanding to guide exploration, and using its generalization to infer better solutions in unexplored areas that can be arbitrarily spatially distant.
> > >
> > > With regard to convergence on other games, we do think an investigation of the convergence of MeVa is in order. However, for this paper we are not concerned so much about the accuracy of MeVa's solutions, as we are about the quality of those solutions. An algorithm that converges exactly to a bad solution is generally worse than one that converges approximately to a good solution (e.g. naive learning finding defection vs LOLA finding tit-for-tat on IPD). MeVa's use of a model introduces approximation errors, and as such will not produce exact solutions, especially not on pathological examples. That said, the game you provide seems like it only has the top-left as a Nash equilibrium? Column 1 dominates column 2. We do not see why this should be a difficult game to solve. Could you elaborate?
> > >
> > > Antisymmetric games are certainly interesting and underexplored in this subfield, but our discussion of the antisymmetry of IMP served chiefly as a warning to practitioners. Several of the prior works in this area (LOLA, COLA, M-FOS) and our own until now made the assumption that all three of these games are symmetric. In the case of M-FOS this has led to a meaningless evaluation being reported. We had the same issue in the original submission, but it has now been resolved.
> > >
> > > Thank you again for your time.
> > >
> > > [1] Fairbank, M. (2014). Value-Gradient Learning. (Unpublished Doctoral thesis, City University London)

---

> > > > ### Comment · Reviewer_1uYc · 2023-11-22
> > > > **Response to Follow Up**
> > > >
> > > > I thank the authors for the follow up response. Regarding exploration, the ablation of an absent exploration scheme does answer my question. However, it raises the question of why add exploration at all if MeVa can perform well in its absence. Considering only matrix games and simple policies excluding neural networks, a sophisticated design strategy for exploration is not required. If it helps, authors should consider removing exploration in order to simplify the MeVa algorithm. As for neural policies, these would require an in-depth evaluation of design choices which can be reserved for future works.
> > > >
> > > > Regarding the provided game, the payoff matrix has a simple structure but it presents vastly different approximate and exact equilibria. The game has an exact Nash equilibrium at (100, 100) but also an $\epsilon$-Nash equilibrium at (1, 1) for $\epsilon = 1$. In my opinion, leveraging an approximation-based method would lead players to the approximate equilibrium which has a significantly lower payoff than the exact equilibrium. This way, approximating the solution may impact convergence of MeVa.

---

> > > > > ### Author Response · Authors · 2023-11-22
> > > > >
> > > > > Thank you, we can see why that game is interesting now. However, we are certain that MeVa will discover (100,100), and once trained, will send all policies there regardless of initialization. You could even make the game [[(100,100),(0,0)],[(0,0),(1,1)]] where the suboptimal equilibrium is not $\epsilon$-Nash but exact Nash, and MeVa agents will learn to move toward (100,100) even if they are initialized near (1,1). This kind of situation appears exactly in the Logistic Game we study: there is a bad equilibrium and a good equilibrium, but to smoothly go from one to the other, one or both players need to pay a cost. MeVa measures the meta-value which accounts for exactly that cost, and decides that it is always worth it to move to the other equilibrium. In a single-player situation, this corresponds to finding global optima from anywhere, which is what interests us in this approach.

---

### Official Review · Reviewer_c4Mz · 2023-11-01

**Soundness:** 2 fair
**Presentation:** 3 good
**Contribution:** 2 fair
**Rating:** 5
**Confidence:** 4

**Summary:**

The paper considers the task of opponent modeling in general sum games. The paper attempts to introduce a LOLA-based method that is able to judge policies over a longer horizon than one-step. The paper introduces a value-based method for a meta-game of optimization. This allows the method to avoid directly modeling the policy space updates for each player. Prior art in LOLA assumes that the opponent is a naive learner and uses a one-step lookahead. The paper implements a self-consistent version of LOLA that relaxes these assumptions. The paper evaluates on small matrix games and a logistic game in order to show how the method may help in scenarios similar to that which LOLA was evaluated in.

**Strengths:**

Opponent modeling is an important domain of multi-agent learning. There is much potential for opponent modeling iff the community is able to create a scalable method. The idea of any LOLA-based paper is to relax some of the assumptions that restrict its scalability. The paper attempts to make a more scalable method through the use of a value-based method that does not rely on policy updates. However, the evaluation is restricted to simple domains.

**Weaknesses:**

The biggest worry with the method is the reliance on the REINFORCE algorithm. The sample complexity required in order to perform this method may be incredibly high once scaled to sufficiently difficult action and state spaces. Though the authors make a note of the limit of this method in terms of learning with neural networks and suggest a direction to help scale their method in the future. This is also a general worry about the LOLA-based works. My question is, does the improvements in the paper lead LOLA-based methods closer to scaling to a feasible solution? If so, how?

Another question is how can this method scale beyond two-player multi-agent settings to general multi-agent scenarios. Even an understanding of how this methodology works in three-player games would be quite an interesting topic.

The results appear to show that M-FOS and MeVa are equivalent in performance. I am confused as to what the takeaway here is. Is it that the performance of MeVa is able to do this without the policy gradient? Why not use the policy gradient as those methods are more scalable?

**Questions:**

“We argue that this throws out the baby with the bathwater” Language like this typically is more confusing than helpful. Consider either explaining the metaphor in the context of the paper or removing this line.

Please see Weaknesses for other questions.

---

> ### Author Response · Authors · 2023-11-14
>
> Thank you for your review. Please see our responses below:
>
> - Reliance on REINFORCE: MeVa does not rely on REINFORCE in principle. We did not use any REINFORCE gradients for the experiments because these games can be differentiated exactly. However, we did note that our $U$ formulation requires a way of estimating $\nabla f$, and REINFORCE is an obvious choice. REINFORCE is in no way required though, as one could also use a model $\hat{f}$ to estimate $f$ and approximate $\nabla f$ by $\nabla \hat{f}$ (this is what the $V$ formulation with $\gamma=0$ would provide).
> - *Do the improvements in the paper lead LOLA-based methods closer to scaling to a feasible solution?* Yes and no. Our method requires significantly more data and computation upfront to learn the value function. However we solve two issues with LOLA that could very well keep it from scaling to neural networks:
>   - LOLA's extrapolation rate $\alpha$ needs to be large. Large learning rates don't work with neural networks. MeVa's discount rate $\gamma$ provides a different way of looking further ahead than just enlarging the learning rate, which all else being equal should be helpful for neural network policies.
>   - LOLA's inconsistency (the assumption of naivety) is a manifestation of the exact issue it sought to address in naive learning (inconsistency due to the assumption of stationarity). In a sense, we feel that MeVa is the logical conclusion of the idea of LOLA.
> - The generalization of our method to more than two agents is straightforward: each agent learns their own meta-value approximation and follows its gradients. How the practical implementation will scale computationally and datawise with the number of agents is another matter, although we have no reason to believe it will scale particularly poorly.
> - Yes, M-FOS and MeVa are equivalent in performance, and if one had to choose, one would choose M-FOS for its scalability. However, M-FOS' performance on the coin game is not stellar, so there is value in exploring a variety of solutions to the same problem.
> One interesting takeaway is that our restriction to local updates did not hurt performance -- MeVa is just as well able to dynamically exploit naive and LOLA learners as M-FOS is.
> - "We argue that this throws out the baby with the bathwater" -- we will remove this line.
>
> Thanks again for your time!

---

> > ### Comment · Reviewer_c4Mz · 2023-11-21
> >
> > Thank you for the clarifications. I will keep my score as is.

---

> > > ### Author Response · Authors · 2023-11-22
> > >
> > > We would like to make sure you did not miss our top-level comment, and our revised submission.
> > > - We address your biggest worry regarding REINFORCE. MeVa does not require REINFORCE, even in general. One can use a parametric model of $f(x)$ to trade variance for bias. Modeling $f(x)$ is like modeling $V(x)$ with $\gamma=0$, a much easier prediction problem than for general $\gamma$.
> > > - We revise the notation to be agnostic to the number of agents, so the application beyond two agents is clear. As for scaling, the complexity of interaction will grow quadratically with the number of players, but that is true for any algorithm on any game. Our main scaling issue is the ability to embed policies, which will only grow linearly (possibly sublinearly in terms of sample complexity for symmetric games where parameters can be shared).
> > >
> > > Thank you!

---

### Official Review · Reviewer_VzC1 · 2023-11-01

**Soundness:** 2 fair
**Presentation:** 2 fair
**Contribution:** 3 good
**Rating:** 5
**Confidence:** 4

**Summary:**

This paper proposes an algorithm that allow agents to model learning processes in the extended future with a value function. This generalizes previous work, namely LOLA, which only models the myopic one-step ahead learning dynamics.
It is pointed out that there are previous works, and other generalizations of LOLA, that also learn meta-value functions.
Unlike other attempts at genearlizing LOLA, the claim is that this algorithm is "self-consistent" meaning that "the algorithm does not assume the opponent is naive and it is aware of its own learning process as well".
Results are shown on logistic and matrix games, showing that meta values outperform LOLA uniformly across the 4 environments and that meta-values tend to be competitive with M-FOS.
Unfortunately, head-to-head results between meta-values and M-FOS are not presented due to computational restrictions.

# Decision

While I like several things about this paper and find the proposed algorithm promising, I think the paper has too many issues to warrant acceptance. I am not sure whether these can be adequately addressed in a rebuttal, but I am tentatively rating the paper below the acceptance threshold.

**Strengths:**

- An overall interesting idea that applies important ideas from meta-learning, single-agent reinforcement learning and multi-agent RL. It is also well motivated by the successes of LOLA and other algorithms that they build on. And the theoretical foundations, while not formally rigorous, are convincing.

**Weaknesses:**

- The paper makes a few statements that are vague and which can be true, but require more specificity. Please see the detailed comments.
- Experimental evaluation seems limited from my point of view, but perhaps this is typical for multi-agent work? The policy spaces are small, and even in these small problems the authors allude to computational concerns for head-to-head results between M-FOS and meta-values. I would also like to see error bars on the performance. Overall, I consider this a weakness, but less so than the first weakness which is the primary factor in my decision.

**Questions:**

- Section 1: (Clarity around Proposed improvements to LOLA): You state that your approach is self-consistent and explain that it "does not assume the opponent is naive and that it is aware of its own learning as well". I find this point unclear and/or redundant.

  In my understanding, the first claim is that the policy convergence of your algorithm does not depend on the opponent following a particular algorithm (specifically, an algorithm that is not learning aware). This is interesting, but can be stated more clearly.

  I think the second claim is that your algorithm is aware of its own learning process. But this is also true of LOLA to a first-order approximation, correct?.

  The second contribution states that other approaches to estimating the meta-value using extrapolation using naive learning. Again, the framing around naive vs non-naive learning is confusing and it may help to state explicitly what this means earlier than section 2.1 if it is a focal point of section 1.

- Section 1 (Clarity around Extrapolation): You refer to the term "extrapolation" but do not define it and I do not think it is common terminology. I assume extrapolation refers to the policy being followed in the bootstrapping step, but I am not sure.


- Section 2.2: You introduce LOLA and a few variants before introducing the proposed algorithm. There seems to be more than a few connections between those previous approaches and the meta-value function approach, and I think the paper would benefit from circling back and relating the proposed algorithm to previous work. The results section, for example, only shows the performance benefit of the proposed approach without a clear demonstration of why or how the extended predictions provided by the value function are beneficial.

- Section 2.3 (M-FOS Description): Contrasting the description of M-FOS here with the earlier one, I do not see how M-FOS can be simultaneously "solving the inconsistency" and "not learning, but acting with learning awareness". If the value function does solve the inconsistency then learning an arbitrary meta-policy from this value function should be seen as "learning with learning awareness" rather than merely acting.

- Section 3 (In place of implicit gradients) At the end of section 3 you remark that once the meta-value function is trained, you can substitute the gradient of the approximation with the implicit gradient of the real meta-value funciton. What is not obvious to me is where the gradient of the implicit meta-value function was used i nthe first place.


- Section 4.1 "This variant is more strongly grounded in the game and helps avoid the detachment from reality that plagues bootstrapped value functions"
  Related to my last point, I do not see the proposed advantage. I do not know what it means for something to be more grounded in a game, nor how this reformulation achieves that.

- Section 4.3 (Exploration): I understand that there is a balance between too much noise for learning and too little noise for exploration. The proposed approach, flipping the sign of the final layer, and the accompanying explanation does not make sense to me. Of course perturbing the output layer would change the behavior of the inner policy, but this seems like a very crude source of noise in comparison to the conventional approach of small gaussian noise added to all parameters.

  I am also confused as to why there are interleaved perturbed and unperturbed rollouts. If the unperturbed rollouts are used to update V, then does that mean the exploration procedure is closer to "exploring starts"

- Section 3 and 4 (Overall): This would all be much clearer if you outlined what exactly is the meta-MDP for the meta-value function. It seems to be not episodic, is that correct?


# Minor Comments
- Section 2: You should explicitly state that you are studying two-player differentiable games in the first sentence for clarity.

- Section 4.1 (Bellman Equation for U): It would be good to show more details how you arrived at this bellman equation as it was not immediately obvious to me (Expanding V(x + \alpha \nabla V(x)) and substiuting U(x) within the recurison). I also do not see why this would make any difference in learning the value function, as something similar can be done in a single agent RL setting. It is not clear what advantage the correction formulation provides.

---

> ### Author Response · Authors · 2023-11-14
>
> Thank you for a very thorough review. We believe several of the issues raised are the result of misunderstandings, we hope to clear them up below.
>
> - Section 1: (Clarity around proposed improvements to LOLA):
>
>   The point is unclear rather than redundant. In the context of LOLA, "consistency" means that the optimization process in the simulated updates is the same as that in the true updates. In LOLA, the simulated update is a naive one, while the true update is a LOLA update. In naive learning, the simulated update is no update at all, while the true update is a naive update.
>
>   LOLA exhibits two kinds of inconsistency:
>   - The simulated update is naive when the true update is a LOLA update.
>   - The simulated optimization process only optimizes the opponent and not the agent itself (equivalently: the agent itself is simulated to not learn at all).
>
>   We swept this second kind under the rug in the name of simplicity: LOLA as described by Eqn (2) only exhibits the first kind of inconsistency, while LOLA proper (given in Appendix B) exhibits both. Reviewer rws5 also appeared to be confused by this, so we will make it explicit, ideally reworking the equations to be written from a single player's perspective like in MetaMAPG. Apologies for the confusion.
>
>   We would moreover like to clarify that consistency in this sense is not related to convergence of the algorithm. It also does not mean invariance to the opponent's algorithm; a consistent surrogate explicitly needs to account for the opponent's algorithm. It may perhaps be construed as an equivariance.
> - Section 1 (Clarity around Extrapolation): Extrapolation is used in the sense of forecasting the future (of the optimization process). We will make sure to clarify this in the paper as it is indeed a pretty overloaded term.
> - Section 2.2: Good point, we will add more discussion for both the Logistic game and the matrix games.
> - Section 2.3 (M-FOS Description): You say "learning an arbitrary meta-policy from this value function should be seen as "learning with learning awareness" rather than merely acting"; we think this mixes up levels of learning. Learning an arbitrary meta-policy would be "learning to act with learning awareness", whereas learning a local meta-policy is "learning to learn with learning awareness". We will clarify this in the paper, as it is a subtle point.
> - Section 3 (In place of implicit gradients): it is used to update the policies $x$ in the inner learning process. We use $V$ as a surrogate for $f$; it is a game on which naive learning is naive-learning-aware.
> - Section 4.1 (Grounding): the practical difference between the $U$ and $V$ formulations is that the $U$ formulation provides useful gradients from the start. Whereas $\nabla V$ relies entirely on the model, $\nabla f + \gamma \nabla U$ sort of "defaults" to the naive gradient $\nabla f$ and learns to modify it eventually. In practice it appears the gradients $\nabla V$ (and $\nabla U$) start out nearly zero, and falling back to the naive gradient allows the system to discover a (minimal) variety of policies. In the process of fitting these policies, the model starts producing more useful gradients.
>
>   One failure mode that we have observed with $V$ is a tendency for policies to become overly cooperative. This is a consequence of the TD error $f(x) + \gamma V(x') - V(x)$ (where $x'=x+\alpha \nabla V(x)$) having minimal weight on the ground truth term $f(x)$ and comparatively large weight (effectively $\frac{\gamma}{1-\gamma}$) on the bootstrapped return estimate $V(x')$. With $V$, the $f(x)$ term in the TD error is the only way in which ground truth enters the system. With $U$, the ground truth additionally enters the system through the use of $\nabla f$ in the optimization process. This is the main sense in which we say $U$ is more strongly grounded in $f$.
>
>   Evidently, we need to discuss this more explicitly in the paper.
> - Section 4.3 (Exploration): There may be a misunderstanding -- we do not apply the noise to the inner policy ($x$) but to the meta-policy ($V$). A perturbation to the meta-policy will cause the meta-rollouts ($x$ optimization trajectories) to go in systematically different directions than they otherwise would. By flipping signs on the learned features in terms of which $V$ predicts the value, we hope to make those directions semantically meaningful. Adding gaussian noise to the parameters of $V$ would potentially do a similar thing, however it is unclear how to scale such noise especially when parameters have an extremely wide range of different sensitivities. Multiplicative noise seems more appropriate to us.
> - Section 3 and 4 (what exactly is the meta-MDP): It is indeed not episodic. We will fully specify the meta-MDP in the paper. Most of the pieces are there in the prose in Sec 2.3, except the action space which we left vague because it differs between the prior work and ours (where it is local and depends on the state).
>
> Thank you!

---

> ### Author Response · Authors · 2023-11-14
>
> We would like to provide an important clarification regarding the final sentence of your summary:
>
> *Unfortunately, head-to-head results between meta-values and M-FOS are not presented due to computational restrictions.*
>
> The restrictions are not computational but methodological; essentially it is not clear how to perform the experiment fairly. It is easy to compare LOLA vs LOLA head-to-head because the meta-policy (LOLA) does not have any parameters to be learned upfront. M-FOS and MeVa *do* have parameters, and comparing M-FOS vs LOLA (or MeVa vs LOLA) involves training M-FOS (or MeVa) with optimization trajectories so they can adapt to LOLA's behavior.
>
> Comparing M-FOS vs MeVa head-to-head requires training them jointly. M-FOS and MeVa learn meta-policies; let us call the process by which they learn these meta-policies ``meta-meta-policies'' (concretely, naive learning). The purpose of a head-to-head evaluation of meta-policies (e.g. LOLA vs LOLA) is to learn something about the interaction between these meta-policies. However, when trained jointly, these meta-policies are moving targets, and whatever we observe will be confounded by the interactions of the meta-meta-policies.
>
> Moreover, we would wish to choose hyperparameters for each method, in order to give both a fair opportunity. Do we choose the ones that make MeVa look good, or the ones that make M-FOS look good? Or do we choose the ones that make both look equally good? At the risk of carrying the terminology too far, we might call the hyperparameter optimization scheme a meta-meta-meta-policy. It's two-player general-sum games all the way up.

---

### Official Review · Reviewer_ZYT4 · 2023-11-01

**Soundness:** 3 good
**Presentation:** 3 good
**Contribution:** 3 good
**Rating:** 5
**Confidence:** 3

**Summary:**

The paper proposes the MeVa method designed for two-player zero-sum meta-games. By extending the concept and form of "looking ahead" methods, and incorporating a discounted sum over returns, it allows for an extrapolated approximation of the meta-value. Additionally, from a practical standpoint, MeVa employs $U(x)$ to approximate the extrapolated value, which helps avoid the detachment from reality often encountered with bootstrapped value functions. Experimentally, the study analyzes the method's behavior on a toy game and makes comparisons to previous work on repeated matrix games.

**Strengths:**

* **Novelty**: This work innovatively extends the concept and form of "looking ahead" methods, while introducing a discounted sum over returns. This allows for an extrapolated approximation of the meta-value, sidestepping the need for approximating the gradient of the policy.
* **Presentations**: The paper presents its viewpoints and theoretical discussions in a lucid and straightforward manner, making it easy for readers to grasp its underlying premise and theoretical implications.
* **Experimental Analysis**: The study compares MeVa with methods like LOLA, HOLA, and COLA. In the meta-games, MeVa demonstrates superior performance.
* The paper fairly analyzes the limitations and prospects of the proposed method.

**Weaknesses:**

* **Theoretical Guarantees**: Although the method leverages the concept of looking ahead, given its use of meta-value, I believe there should be some theoretical analysis regarding its convergence and computational complexity, etc. However, I did not find such discussions in the paper.
* **Experiments**: The experiments primarily compare against baseline methods that are not specifically designed for meta scenarios. I believe it would be more informative to design experiments comparing with other meta-specific methods beyond M-FOS, such as Meta-PG, meta-MAPG.
* **Reproducibility**: The source code is not submitted, making reproducibility uncertain.
* **The targeted scenarios are somewhat restrictive**:
    * The scope of the approach seems somewhat limited. As it's currently tailored for two-player zero-sum meta-games, it might be challenging to expand to tasks with more complex state and action spaces.
    * The paper assumes that one can observe the strategy parameters of the opponent. This assumption might be difficult to uphold in real-world tasks.

**Questions:**

* As previously mentioned, within the MeVa framework, is it possible to analyze the algorithm's convergence properties?
* Please include benchmark experiments for Meta-PG and Meta-MAPG to provide a more comprehensive comparison.
* In two-player zero-sum games, might some baseline methods that compute equilibria also be included in the comparison experiments?

---

> ### Author Response · Authors · 2023-11-13
>
> Thank you for your review. Please see our responses below.
>
> - Theoretical Guarantees: we do briefly discuss convergence at the end of Section 3.2. We also expected to be able to inherit convergence results from Q-learning. Unfortunately, multi-agent Q-learning is not known to converge even in the discrete/tabular case. The Bellman operator is generally not a contractive map when multiple agents maximize different objectives.
> - Experiments: we will include a comparison to MetaMAPG on the matrix games. This will take a few days to come together. The M-FOS authors (Sec 5.1) note that MetaMAPG does not scale beyond 7 updates, and use a variant based on exact gradients. We will follow them in this regard.
> - Reproducibility: apologies, we should have submitted the code here on OpenReview. It is available at https://github.com/MetaValueLearning/MetaValueLearning .
> - The targeted scenarios are somewhat restrictive:
>   - We present the method in two-player form for simplicity, but the generalization to n players is straightforward: each player learns and optimizes their own meta-value.
>   - We are not sure why you say the method is tailored to zero-sum meta-games; that does not appear to us to be the case. E.g. on IPD, some meta-policies lead to defection whereas others lead to cooperation, with different total payoffs.
>   - It is true that we assume access to opponent parameters. We believe working on top of opponent models is entirely possible, but the science is better done in a controlled setting. Even there, the nonstationarity problem of MARL is far from solved.
> - *In two-player zero-sum games, might some baseline methods that compute equilibria also be included in the comparison experiments?*
> The only zero-sum game we consider is Iterated Matching Pennies, and it only has a single Nash equilibrium at the uniform policy pair. Naive learning finds it without issue, so it's unclear what these other algorithms would add. Could you elaborate?
>
> We hope to have addressed some of your concerns. Thank you!

---

### Official Review · Reviewer_rsw5 · 2023-11-03

**Soundness:** 3 good
**Presentation:** 2 fair
**Contribution:** 3 good
**Rating:** 6
**Confidence:** 2

**Summary:**

The paper proposes Meta-Value Learning (MeVa), a general framework for learning with learning opponent awareness in MARL. MeVa uses a meta-value method to account for longer-term behaviours of opponents and does not require policy gradients. It is consistent and far-sighted, avoiding the need to explicitly represent the continuous action space of policy updates. Many evaluations are conducted on various games, including the Logistic Game, Iterated Prisoner's Dilemma, Iterated Matching Pennies, and the Chicken Game, demonstrating MeVa's effectiveness in opponent shaping and cooperation for MARL. The method shows its merit in achieving cooperation where self-interest warrants it, without being exploitable by other self-interested agents.

**Strengths:**

**Originality:**

MeVa is a novel method based on meta learning for dynamic opponent modelling in MARL. Unlike previous methods, MeVa is consistent, meaning it does not assume its opponent is naive and is aware of its own learning process. MeVa is far-sighted, i.e., it looks more than one step ahead through a discounted sum formulation, which allows it to account for longer-term and higher-order interactions.

**Quality:**

Overall, MeVa is a high-quality method. It extends previous LOLA and other methods with value learning, which is based on value learning and does not require policy gradients anywhere. MeVa can be applied to optimization problems with a single objective.

**Clarity:**

Overall, the writing is good. It introduces comprehensive background and related works, which is easy for readers to follow.

**Significance:**

MeVa brings new insights into the MARL community, including its consistency on learning the full dynamics of the other agents, far-sightedness, value learning and implicit Q-function.

**Weaknesses:**

1. Scalability: The main weakness of MeVa is its scalability, particularly when learning in environments with more agents. The method may struggle to handle large parameter vectors and complex multi-agent interaction when more agents are involved.

2. Writing in the methodology section: This section can be improved by using similar notations from previous works, such as LOLA, Meta-PG and Meta-MAPG. It will make it consistency in notation and easy to follow.

3. Algorithm 1 is hard to follow. It would be great to add more explanations.

**Questions:**

Q1: In page 3,  authors mentioned that “nevertheless there is always a gap where each player assumes it looks one step further ahead than their opponent”. Could you please explain why there is a gap in LOLA?

Q2: Why is Equation (3) a better surrogate?

---

> ### Author Response · Authors · 2023-11-13
>
> Thank you for your review. We respond to the issues raised below.
>
> Weaknesses:
> 1. It is true that the complexity of the problem increases with the number of agents, however it is not clear that this is a weakness of our method, as opposed to an inherent fact of life in MARL.
> 2. Another reviewer also raised a concern about the clarity of the notation in Section 2. We will revisit the notation and ensure the meta-MDP is fully defined.
> 3. The current algorithm is hard to follow because it was written to have full details regarding the application of the practical techniques. We will move this to an appendix and provide a simplified, readable presentation in the main text.
>
> Questions:
> 1. Our words were confusing and we will rewrite them. To see the gap, consider two LOLA agents A and B training against each other. A assumes B uses naive learning, when in truth B uses LOLA as well. And similarly, B assumes A uses naive learning, when A actually uses LOLA. In a sense, both players think they are smarter than their opponent. This is the gap. The same holds in naive learning, where each agent assumes the other to not be learning at all. And the same holds in HOLA2 when A assumes B uses LOLA and B assumes A uses LOLA, when in fact they are both using HOLA2.
> 2. Two agents that follow the gradient of (3) make the correct assumption about how optimization would proceed, namely that both agents follow the gradient of (3). (Our notation in Section 2 is somewhat simplified and does not reflect LOLA's original proposal to only imagine the opponent's update; instead we consider the agents to be aware of their own learning as well. Ignoring the agent's own update would be a different, unrelated source of inconsistency.)
>
> We hope that clears things up.

---

### Official Review · Reviewer_zqYk · 2023-12-07

**Soundness:** 3 good
**Presentation:** 3 good
**Contribution:** 3 good
**Rating:** 5
**Confidence:** 5

**Summary:**

I was asked to give a last minute review. I understand the authors will not have a chance to respond, which is not ideal; I hope to mostly bring up points that are touched on by existing reviews.

The authors introduce MeVa, a method which is a consistent and far-sighted method for opponent-shaping. MeVa works by applying Q-learning/DDPG-like method to the meta-game.

**Strengths:**

Originality:

- Q-Learning-based methods have not been applied to the meta-game in opponent-shaping scenarios.

Quality:

- The experiments are thorough.

Clarity:

- The paper is clearly written

Significance:

- Opponent shaping is becoming increasingly important as more real-world AI systems are deployed

**Weaknesses:**

- The paper makes heavy comparisons to meta-learning based methods, but notably completely omits them from the introduction (which is quite important for understanding the contributions of this paper). For example, when factoring in M-FOS's existence into the contributions and claims made in the Introduction, the main one is that MeVa is value-based and not policy-gradient-based. I'm not sure if this counts as a significant contribution. By omitting these methods from the introduction, the paper makes it seem like solving LOLA's inconsistency and myopia are novel contributions.

    - This is largely related to points brought up by Reviewers c4Mz and 1uYc.

- I don't understand the difference between "learning with opponent learning awareness" and "acting with opponent learning awareness" the authors mention in Section 2.3. The authors seem to be implying that learning = gradients. Under this distinction, "in-context learning" is "in-context acting", which I don't think most researchers would agree with.

    - This is related to points brought up by Reviewer VzC1

    - I'm assuming MeVa uses fast policy changes in IMP, which seems to defeat the whole point of the "gradual dynamics that are characteristic of learning".

I concede that these concerns could be alleviated with some simple discussion or minor modifications. However, I do not feel strongly positive enough about the paper to overrule the existing reviews.

**Questions:**

- Appendix D raises an interesting point; however, I'm surprised it makes a difference for Matching Pennies. Since M-MAML only defines an initialization and from there performs NL, I can't imagine what initializations would allow you to outperform a uniformly-sampled one on average. I would assume there is some symmetry in the meta-game policies (much like there is a symmetry in the underlying game) [NOTE: I am not using symmetry the same way here as the authors do in the paper. I am merely stating that there exists a one-to-one mapping]. I'm curious what initializations M-MAML learns.

---

### Author Response · Authors · 2023-11-21
**Revision with expanded experiments**

Dear reviewers,

Thank you all for your patience while we addressed the issues with our experiments. We have uploaded a revision with the following modifications:
- The notation is written from a single player's perspective, similar to how it is done in the MetaMAPG and M-FOS papers. This makes it clear how to apply MeVa in the case where the opponent meta-policies are unknown. We still make use of the simultaneous/vector notation when discussing the case where all players use the same meta-policy (Sec 2.2).
- We have moved the detailed algorithm to the appendix, and introduced two simple algorithms: one for the case where all players use MeVa, and one for the case of unknown opponent meta-policies.
- We have clarified the definition of the meta-game. In doing so, we discovered an error in our argument relating MeVa to Q-learning. Specifically we glossed over the expectation over the next state when defining the Q-function in terms of the V-function. We have now made the argument much more precise.
- We have made the derivation of the U bellman equation explicit, and go into a little bit more detail on the practical implications of the reformulation.
- The matrix game tournament now includes M-MAML, an exact-gradients special case of MetaMAPG. While we were trying to make sense of the results, we discovered a bug that is also present in M-FOS' code. Essentially, the games are assumed to be symmetric ($f_1(x_1,x_2)=f_2(x_2,x_1)$), but Iterated Matching Pennies is not symmetric. This requires training everything twice, to have a model for each side of the game. We go into detail on the issue in Appendix D. It is the main reason we took so long to update the submission.
  Unfortunately we were not able to report the M-FOS vs M-MAML comparison as we forgot to run it, however even if we did it would have been subject to the above bug in the M-FOS code. We will include this comparison in the camera-ready.
- We have tweaked the ablation experiment, mainly to make the "no exploration" case fall back to a reasonable default exploration. Some of you suggested (meta)action noise, and some suggested (meta)policy parameter noise. Ultimately we decided on Noisy Networks (Fortunato et al, 2017), as it is the choice of the Rainbow authors as well. With some tweaking to the initialization (1e-4 as the initial standard deviation worked better than 0.017), this basic exploration scheme works pretty much as well as our sign flipping scheme. If the reviewers desire it, we may abandon our scheme for the camera-ready, and use Noisy Networks exclusively.
- We have provided the (Colab) notebook for the matrix game experiments (tournament and ablation) in the supplementary materials.

With your feedback, we were able to greatly strengthen the paper. We ask that you please look at our revision and let us know if your concerns have been resolved. Thank you for your time.

---

### Meta-Review · Area_Chair_qRUk · 2023-12-07

**Metareview:**

This paper proposes a value-based meta-learning framework for opponent shaping in general-sum games.

Strengths:
- Opponent shaping is very timely
- The experimental evaluation of the method is thorough
- The method is well motivated

Weaknesses:
- The authors fail to properly motivate the method in the context of existing work, such as M-FOS. Specifically, a number of contributions / claims in the introduction were already done in M-FOS rather than being novel in this work (addressing inconsistency, long term shaping).
- Novelty is somewhat limited compared to the claims. This is essentially a value-based version of M-FOS with smart implementation choices / approximations.
- Some reviewers struggled to fully understand the paper, suggesting the the writing could be further improved.

While based on my own assessment I'd be happy to accept the paper if there is space at the conference, I also want to do justice to the reviewing process and believe that the paper could be further improved for a future venue.

**Justification For Why Not Higher Score:**

This is a borderline paper, but I believe that the writing / presentation of the claims could be clarified for a future venue. This would both make it easier to understand the contributions and help other contextualise the work within the literature.

**Justification For Why Not Lower Score:**

N/A

---

### Decision · Program_Chairs · 2024-01-16

Reject